Corrected: Author correction

# Vocal state change through laryngeal development

Yisi S. Zhang[1], Daniel Y. Takahashi[1], Diana A. Liao[1], Asif A. Ghazanfar [1,2,3,5]* & Coen P.H. Elemans [4,5]*

Across vertebrates, progressive changes in vocal behavior during postnatal development are typically attributed solely to developing neural circuits. How the changing body influences vocal development remains unknown. Here we show that state changes in the contact vocalizations of infant marmoset monkeys, which transition from noisy, low frequency cries to tonal, higher pitched vocalizations in adults, are caused partially by laryngeal development. Combining analyses of natural vocalizations, motorized excised larynx experiments, tensile material tests and high-speed imaging, we show that vocal state transition occurs via a sound source switch from vocal folds to apical vocal membranes, producing louder vocalizations with higher efficiency. We show with an empirically based model of descending motor control how neural circuits could interact with changing laryngeal dynamics, leading to adaptive vocal development. Our results emphasize the importance of embodied approaches to vocal development, where exploiting biomechanical consequences of changing material properties can simplify motor control, reducing the computational load on the developing brain.

[1] Princeton Neuroscience Institute, Princeton University, Princeton, NJ 08544, USA. [2] Department of Psychology, Princeton University, Princeton, NJ 08544, USA. [3] Department of Ecology & Evolutionary Biology, Princeton University, Princeton, NJ 08544, USA. [4] Department of Biology, University of Southern Denmark, 5230 Odense M, Denmark. [5] These authors jointly supervised this work: Asif A. Ghazanfar, Coen P.H. Elemans. *email: asifg@princeton.edu; coen@biology.sdu.dk

Vocal development is marked by progressive changes in behaviors. The development of human speech, for example, begins with crying and protophones, continues through babbling and the production of speech-like words to, eventually, complex sentences[1]. The brainstem central pattern generators (CPGs) encoding vocal motor output are well-documented brain areas in vertebrates (birds, bats, fish, and frogs)[2], including monkeys[3] and humans[4], and their modulation by higher-order brain areas is also known[5,6]. However, can these circuits fully account for the dramatic changes in infant vocal output?

From 3 to 9 months of age, human infant vocalizations, transitioning from cries to cooing sounds, change in their fundamental frequencies and exhibit signatures of nonlinear dynamics such as noisiness and subharmonics[7]. Later, infants can produce sounds without these chaotic elements, and the ability to do so has long been hypothesized to be due solely to increased neural control of the vocal organ[7–9]. However, the larynx itself is also changing during postnatal development[10–15]. Thus, an alternative hypothesis (though, not mutually exclusive) is that changes in infant vocalizations are related to changing laryngeal dynamics. Such hypotheses are difficult to test in humans. We thus lack experimental evidence that causally relates changes in vocal output with laryngeal change as a function of growth and development.

We used infant marmoset monkeys as a model to test the role of the changing larynx in the development of vocalizations. Like human infants[16], infant marmosets at the same life history stage spontaneously produce sequences of immature and mature vocalizations. Over the course of ~2 months, infant vocalizations exhibit changes in their acoustic properties that reflect a transition from producing mostly immature-sounding contact calls (i.e., cries) to mature contact calls (i.e., phees)[17,18]. This transformation of cries into adult-like contact calls includes (as they do in human infants[7]) changes in their fundamental frequencies as well as signatures of nonlinear dynamics such as noisiness and subharmonics; these latter elements of instability disappear as the infants get older[17,19]. We tested the hypothesis that the transition from cry-like states to mature-like states in the marmoset monkey vocal repertoire is due to the changing biomechanical properties of the developing larynx.

We show that vocal state changes in infant marmoset monkeys are in part the adaptive consequences of the gradually changing laryngeal properties. A sound source switch from the vocal fold (VF) proper to the apical vocal membrane (VM) causes the transition from noisy, low-frequency cries to tonal, higher pitched contact vocalizations. Because of its increased VM stiffness, the adult larynx can only produce loud, higher pitched vocalizations produced with higher efficiency. Furthermore, using an empirically based computational model of descending motor control, we show how neural circuits can interact with changing laryngeal dynamics, leading to adaptive vocal development. Our results emphasize the need for an embodied framework of vocal motor control, especially during postnatal development where the brain must coordinate its activity with a rapidly changing body[19,21].

## Results

**Contact call transitions from two attractor states to one**. Our first step was to quantitatively characterize the state changes in marmoset monkey vocal development by measuring the probability for the production of certain acoustic features. The vocalizations were collected in a controlled, undirected context, which reliably elicits contact vocalizations[17,21]. We recorded infant (immature and mature contact calls, $n = 6576$ vocalizations, $N = 10$) and adult contact calls ($n = 887$ vocalizations, $N = 5$) (Fig. 1).

The spectral features of marmoset vocalizations are well described by their fundamental frequency ($f_o$) and Wiener entropy (WE), a measure of the richness of the power spectra[17]. The immature and mature contact calls produced by infants form distinct clusters in the acoustic space: mature sounding 'phee' calls have high $f_o$ (mean ± s.d.: $8.93 ± 0.68$ kHz) and low WE ($-34 ± 4$ dB), while the immature 'cry' calls have low $f_o$ ($640 ± 60$ Hz) and high WE ($-15 ± 3$ dB) (Fig. 1b). In contrast, the adults produce exclusively phee vocalizations that cluster at a high $f_o$ ($7.4 ± 0.3$ kHz) and low WE ($-35 ± 4$ dB). To map acoustic feature probability, we calculated $f_o$ and WE on a 10-ms basis. The infant vocal acoustics form two distinct peaks in the parameter space, or two attractor states. We define an attractor state as a region with a high probability density in the acoustic space. One attractor centered around $f_o = 600$ Hz and WE $= -16$ dB, and the other centered around $f_o = 9$ kHz and WE $= -37$ dB. These attractors correspond directly to the immature and mature contact calls, respectively (Fig. 1e). In adults, we identified one attractor around $f_o = 7$ kHz and WE $= -37$ dB, corresponding solely to the mature contact phee call (Fig. 1f). Thus, over the course of normal vocal development, marmoset contact vocalizations progress from two attractor states to one.

**The infant larynx has a greater acoustic capacity**. To test if these attractors result from constraints in laryngeal dynamics that change over development, we investigated the vibratory behavior of excised larynges from infant ($N = 3$) and adult ($N = 4$) marmosets. Our setup allowed us to carefully and independently control both VF tension and subglottal pressure (see Methods). Our first specific hypothesis was that the difference between infant and adult vocalizations was that the amount of subglottal pressure needed to induce VF vibration is lower in infants than in adults. This hypothesis is derived from biomechanical models that simulate vocal production using two time-varying, lumped parameters—the subglottal pressure and laryngeal tension[19,22,23]. These models all posit that $f_o$ is positively correlated with subglottal pressure and VF tension. Thus, we wanted to know if infants have a lower $f_o$ because the minimum pressure needed for the infant larynx to produce sounds (aka the phonation threshold pressure, PTP) is lower than for adult larynges (Fig. 2a). We induced phonation in excised larynges of infants and adults, precisely controlling subglottal pressure and VF length. We quantified the PTP of vocalizations by applying subglottal pressure ramps from 0 to 3 kPa at 1 kPa s$^{-1}$. Our results falsify the model hypothesis by revealing that the PTP was significantly higher in the larynges of infants ($2.05 ± 0.37$ kPa) versus adults ($0.62 ± 0.13$ kPa) ($p < 0.001$, linear mixed-effects model) (Fig. 2c). These results show that the production of immature calls—the cry attractor state—by infant larynges cannot be explained by a lower PTP.

An alternative hypothesis is one of capacity, that the adult larynx simply cannot produce the cry attractor state like the infant larynx. That is, cry production is outside the available range of the adult vocal control space (Fig. 2a, right panel). If the capacity hypothesis is true, then the variation of acoustic features driven by the same change of control parameters should be greater for the infant larynx. To test this hypothesis, we applied sweeps through the vocal control parameter space by independently varying VF strain and subglottal pressure (Fig. 2d–g). The VF strain was calculated as the fractional change of VF length with respect to the resting length. The mean resting lengths of infant and adult VFs were $1.4 ± 0.2$ mm and $2.8 ± 0.1$ mm, respectively. In the infant larynx, sounds produced at low VF strain were very similar to the infant marmoset cry (Fig. 2d, left panel; Supplementary Fig. 1). At a specific VF length change

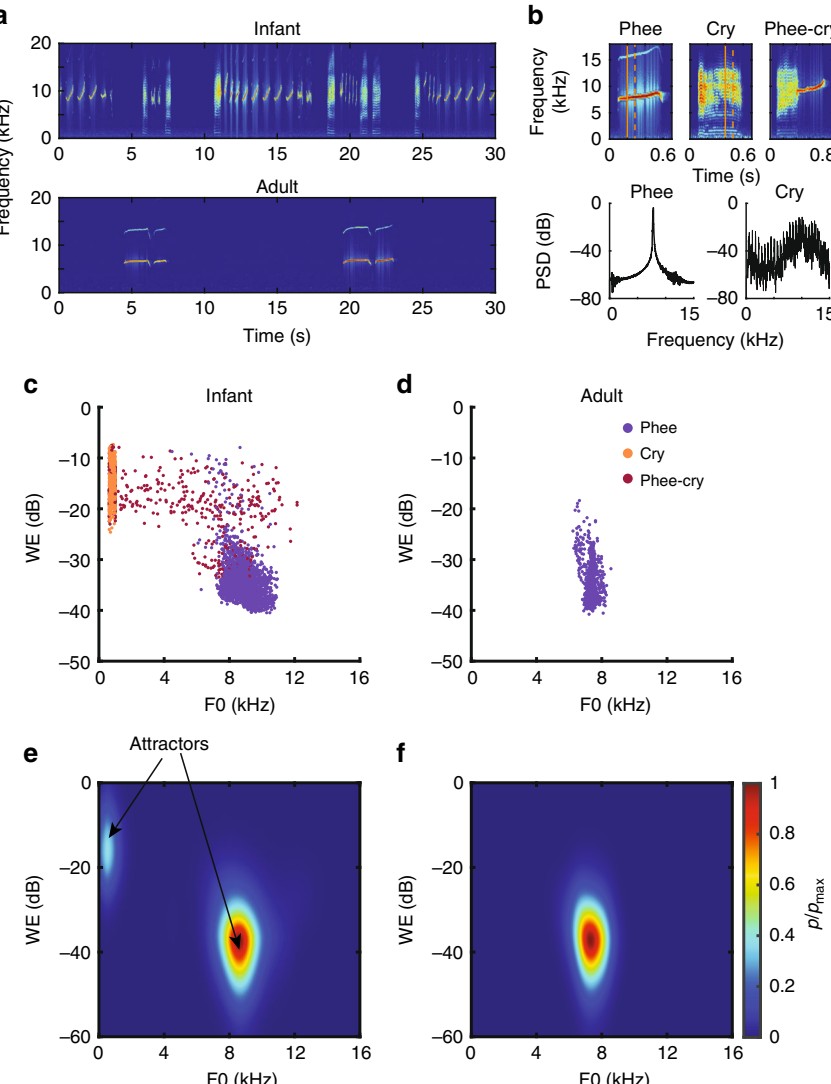

**Fig. 1** Infant and adult marmoset vocalizations fall into different attractors in the acoustic space. **a** Spectrograms of infant (top) and adult (bottom) marmoset undirected vocalizations. **b** Exemplars of three different contact calls: Phee, Cry, and Phee-cry. Vertical lines in the spectrogram indicate the segment used for calculating the power spectral density (PSD) below. **c**, **d** The acoustic features of infant and adult contact calls in the Wiener entropy (WE)–$f_o$ space. **e**, **f** The max-scaled probability density of infant and adult vocal acoustic features in the WE–$f_o$ space. Source data are provided as a Source Data file.

corresponding to 17–45% strain, we observed a sharp transition in the $f_o$ in each individual from ~2 to ~9 kHz, the latter corresponding well to the mature-sounding contact call (Fig. 2d, left panel). The sound pressure phase portrait suggests a transition between two distinct vibratory modes (Supplementary Fig. 2). In contrast, in the adult, $f_o$ transitions were never observed (Fig. 2d, right panel). The different behaviors of the infant and adult larynges yielded distinct $f_o$ maps in the pressure–strain space, in which a steep $f_o$ transition exists in the infant map that is absent in the adult map (Fig. 2f, g). At very low strain and high pressure (>5 kPa), both infant and adult larynges entered a chaotic acoustic region featured by high WE value that did not correspond to any natural marmoset monkey calls. We also assessed the contribution of control parameters to the variation of acoustic features and found that $f_o$ control is redundant with respect to pressure: $f_o$ is primarily driven by VF length. Infant larynges yielded greater variation than adult larynges in the control space for both $f_o$ (Fig. 2h; $p \ll 0.001$, test for slope equal to 1) and WE (Fig. 2i; $p < 0.001$, test for slope equal to 1). These

findings support the hypothesis that the cry attractor state is outside the vocal capacity of the adult larynx.

**Changing VF material properties**. VF material properties are a likely candidate to induce laryngeal changes at developmental timescales, but we lack the knowledge of how VF material properties could change over time. While VF vibratory kinematics, and laryngeal acoustic output, results from the complex interplay of viscoelastic properties, fluid flow, and acoustics[24–26], modeling the VF as a 1-dimensional vibrating string can, at a first approximation, partially explain $f_o$ ranges observed across a variety of species, even after accounting for body size[24]. Such string models predict that $f_o$ is proportional to tissue stress over density and inversely proportional to VF length[24]. VF stress typically increases nonlinearly with strain, which allows an extension of $f_o$ by two different mechanisms. First, as the maximum VF length is constrained by laryngeal geometry, nonlinearly increasing VF stress with strain allows an upward extension of $f_o$ range. Second, any nonlinearity in the dynamical

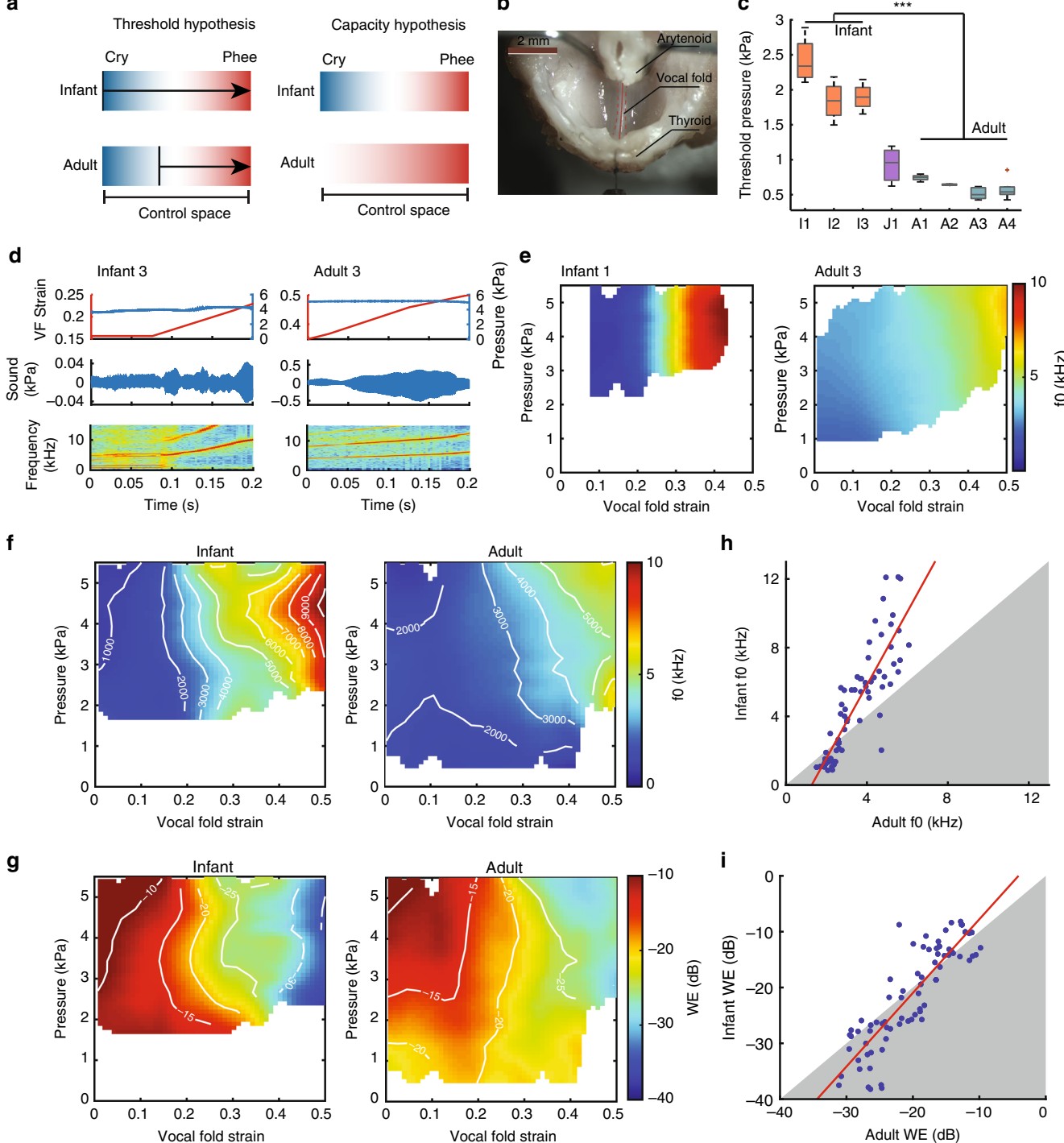

**Fig. 2** The infant larynx has greater vocal capacity than the adult larynx. **a** Two alternative hypotheses for infant cry production. Threshold hypothesis: both the infant and the adult larynges have the same vocal range, but the adult larynx starts phonation at a higher threshold. Capacity hypothesis: the infant larynx has greater capacity than the adult larynx. Color indicates $f_o$. **b** Excised larynx setup: an infant marmoset larynx mounted on an artificial trachea with arytenoid cartilage adducted and the position of thyroid cartilage manipulated by the lever arm of an ergometer. **c** Comparison of phonation threshold pressure between infant and adult excised larynges. **d** Exemplars of sound generated from an infant and an adult larynx while the vocal folds were stretched at fixed pressures. From top to bottom: the real-time vocal fold strain and subglottal pressure, sound pressure, and sound spectrogram. **e** $f_o$ maps of an infant and an adult larynx in the control space of pressure vs. vocal fold strain. **f, g** Mean $f_o$ maps (**f**) and mean WE maps (**g**) of infant ($n = 3$) and adult ($n = 4$) larynges. Iso-$f_o$ contours (in Hz) and iso-WE contours (in dB) are overlaid, showing that the pressure control is redundant. **h, i** Comparisons between the infant and adult $f_o$s (**h**) and WEs (**i**) generated at the same pressure and vocal fold strain settings. Slopes significantly greater than 1 indicate greater capacity of the infant larynx than the adult larynx. Source data are provided as a Source Data file.

system that sets the VF vibration kinematics can lead to multiple discrete and stable attractor states of VF vibratory patterns[27].

We measured the stress–strain properties of VF tissues in infant and adult larynges (see Methods). A salient part of the VF geometry in some species are thin apical extensions called VMs[28,29]. Their function in vocal production remains untested experimentally but it has been suggested that they act as low mass oscillators that can vibrate almost independently of the VF proper[30], and thus support, for instance, the production of high-frequency vocalizations[29,31,32]. We therefore quantified the stiffness of VF proper and VM separately. The stress–strain curves were highly nonlinear and could be fitted with combined linear and exponential models at low and high strain regions[33,34] (Fig. 3b). The linear elastic limit was around 12–20% strain (fitted parameters summarized in Table 1). Overall, for both infant and adult larynges, as the strain increased, the VM stress increased more rapidly than the VF stress ($p < 0.001$, linear model). Following the string model, at a strain of 50%, both the infant and adult VMs can yield a 3–4 times higher $f_o$ than the VF. Consistent with predictions for the role of the VM, this would enable high-frequency vocalizations at a greater efficiency[30]. However, it does not explain the $f_o$ magnitude difference between the observed immature and mature call attractors. The stress–strain responses between infant and adult larynges were significantly different between their VMs ($p \ll 0.001$, linear model), but not between their VFs ($p = 0.42$, linear model). Thus, the maturation of larynx is driven at least in part by a viscoelastic change in the VM. The VM trajectories (Fig. 3b) show that in low strain regions, the stress of the infant VM only starts deviating from the VF at ~17% up to ~45% strain, much later in the trajectory than that of the adult VM (3.9% strain; $p < 0.001$, bootstrap). The infant VM exhibited a similar elasticity as its VF at a low strain region. The divergent point around 17% also coincides with the strain where the frequency jump occurred in the phonation tests of the infant larynx. In contrast, because the stress–strain responses of the adult VM and VF diverge almost immediately; their vibratory modes become distinct very quickly.

**Shift to VM vibrations in adult contact calls**. We speculated that it is this divergence in the elastic properties between the VF proper and the VM that accounts for a switch between distinct vibratory modes. To test this, we quantified glottal dynamics during sound production using a high-speed camera (Fig. 3c–f). During the sub-1-kHz sound produced by the excised infant larynx at low strain (<15%, i.e., relaxed state), we observed vibration with the caudal edges (VF proper) always closing prior to the cranial edges (the VM) (Fig. 3c, top two panels; Supplementary Movie 1). This phase difference along the caudo-cranial axis reflects energy transfer through the tissue[35,36]. Sound excitation occurred during glottal opening and closing (Fig. 3c, bottom panel). However, when the VF and VM were stretched (strain >15%, i.e., stretched state), glottal area dynamics were defined solely by the vibration of the VM (Fig. 3d; Supplementary Movie 2); no underlying VF vibration could be observed. Thus, the rapid transition from low to high $f_o$ was caused by a distinct shift of sound production from the VF to the VM. This transition was accompanied by a shift from relatively complex to regular vibratory patterns (Fig. 3f). In the adult larynges, glottal area dynamics was set exclusively by regular vibration of the VM (Fig. 3e, f; Supplementary Movie 3), which is consistent with our finding that VM stress deviated from VF stress at a low strain in the adult larynges (Fig. 3b).

**The adult larynx is more vocally efficient**. Our data show that the function of the VMs—a morphological innovation observed

in New World monkeys, cats, and bats—is to (i) increase the $f_o$ range and (ii) decrease PTP, corroborating previous theoretical predictions[22,30,37]. Because decreased PTP is indicative of increased vocal efficiency (leading to higher amplitude vibrations for a given lung pressure)[30], vocalizations could theoretically be produced louder with less energy, thereby extending their communicative range. To test this, we calculated the sound source level (SL) as a function of pressure and VF (VF proper and VM) strain. We found that the SL of the emitted sound was positively correlated with pressure as well as strain in both infant and adult larynges ($p < 0.001$ for slopes of multiple linear regression; Fig. 4a). This is consistent with the prediction that the VM—the source oscillator at the higher strain levels—is more energy efficient than the VF proper. Overall, the adult larynx produced sound about 15 dB louder than the infant larynx across all sets of parameters (Fig. 4a). These findings from the excised larynx preparation are consistent with what is observed in naturally produced vocalizations (Fig. 4b). The infant's mature-sounding contact calls are generally louder than its cries, and the adult contact calls are louder than infant's contact calls ($p \ll 0.001$, ANOVA). Assuming sound transmission loss by spherical spreading[38], the 15 dB loudness increase of the contact call from infant to adult results in a roughly six times larger communicative distance. To produce louder sounds, the adult larynx may simply draw more mechanical energy or it might be more efficient in converting mechanical energy to sound. To test this, we estimated its mechanical efficiency (ME): the fraction of the aerodynamic power that was converted to acoustic power[39]. We found that the adult larynx was more efficient at all parameter settings (Fig. 4c), demonstrating that ME increases over development. In addition, the ME was positively correlated with strain for both infant and adult larynges ($p \ll 0.001$ for slopes of multiple linear regression; Fig. 4d). Thus, it is more energy efficient to produce high $f_o$ sounds when energetic constraints allow for them.

**Synchronization of laryngeal and respiratory output**. Our findings do not address the role of the concurrently changing nervous system—the vocal CPGs[3,4] and descending control[6]. In the case of marmoset monkeys, this is important because infants raised without parental care continue to produce noisy, cry-like vocalizations well beyond the age when such calls should have disappeared from the vocal repertoire[40,41]. These animals are not stunted in their bodily growth and it is unlikely that their laryngeal development is dependent upon parental care. It is therefore logical to infer a role for experience-dependent neural control.

To define a neuromechanical framework for embodied vocal motor control, we reverse-engineered the potential real-time control parameters—pressure and strain—that change over development (Fig. 5a). This was possible because the control space mapped from excised larynges matched the range of in vivo marmoset vocalizations (Fig. 2f, g). The probability density maps of the control parameters show two and one attractors present in the infant and adult control space, respectively (Fig. 5b). For infants, the two attractors occur at 10 and 45% strain at subglottal pressure ~3 kPa (Fig. 5b, left panel). These attractors correspond to the production of immature and mature calls, respectively. For adult vocalizations, the attractor at 45% strain and ~5 kPa pressure represents the production of mature contact calls (Fig. 5b, right panel). Thus, in addition to the inability to produce vocalizations with a $f_o < 1$ kHz, adults also do not produce contact vocalizations at low strain.

When these two control parameters are in phase, mature sounding contact calls are produced in both infant and adults (Fig. 5c). Infant marmosets, however, produced immature cry

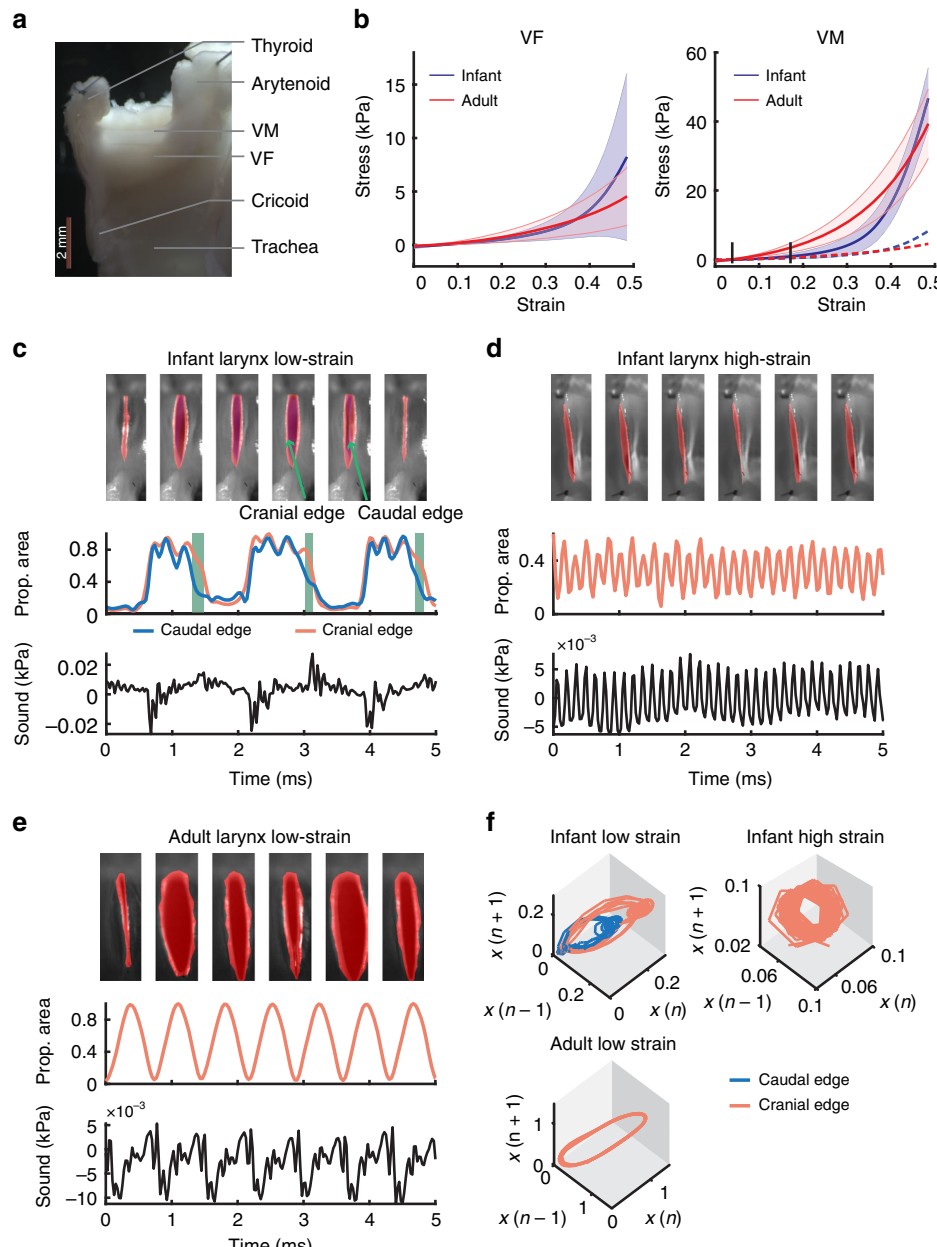

**Fig. 3** Cry production is unique to infant larynx. **a** The sagittal section of an isolated adult marmoset larynx. The relative locations of vocal membrane (VM) and vocal fold (VF) are indicated. **b** Stress–strain responses for infant and adult VF and VM. The black line indicates diverging point between the VM and VF stress along the strain axis. The dashed curves are the VF stress–strain response. The shaded areas are standard deviation. **c–e** Glottal dynamics of **c** an infant larynx at low-strain region, **d** an infant larynx at high-strain region, and **e** an adult larynx at low-strain region. Top to bottom: snapshots of glottal opening and closure with the areas bounded by the cranial (in red; **c–e**) and the caudal (in blue; only **c**) edges, the proportional glottal areas (with respect to the maximum glottal area), and the distance-corrected sound pressure. In **c**, the green bars indicate that the caudal edge closed prior to the cranial edge. **f** The trajectories of the absolute glottal area (in $mm^2$) in the phase space for the instances corresponding to **c–e**.

| Table 1 Stress–strain response fitting parameters | | | | |
|---|---|---|---|---|
| | **a (kPa)** | **b (kPa)** | **A (kPa)** | **B (kPa$^{-1}$)** | **$\epsilon_0$ (%)** |
|---|---|---|---|---|---|
| Infant VF | −0.19 ± 0.21 | 3.13 ± 2.15 | 0.10 ± 0.08 | 8.73 ± 3.75 | 18.4 ± 6.4 |
| Adult VF | −0.18 ± 0.13 | 4.19 ± 2.97 | 0.27 ± 0.18 | 5.94 ± 0.19 | 21.1 ± 0.7 |
| Infant VM | −0.31 ± 0.30 | 5.77 ± 5.33 | 0.15 ± 0.21 | 13.65 ± 3.34 | 13.1 ± 2.6 |
| Adult VM | −0.83 ± 0.52 | 22.71 ± 10.77 | 1.26 ± 0.66 | 7.23 ± 0.57 | 17.4 ± 1.7 |

The stress–strain response was modeled with a linear model at low strain and an exponential model at high strain: $\sigma = \begin{cases} a + b\epsilon, & \epsilon \leq \epsilon_0 \\ Ae^{B\epsilon}, & \epsilon > \epsilon_0 \end{cases}$, where $\epsilon_0$ is the linear limit

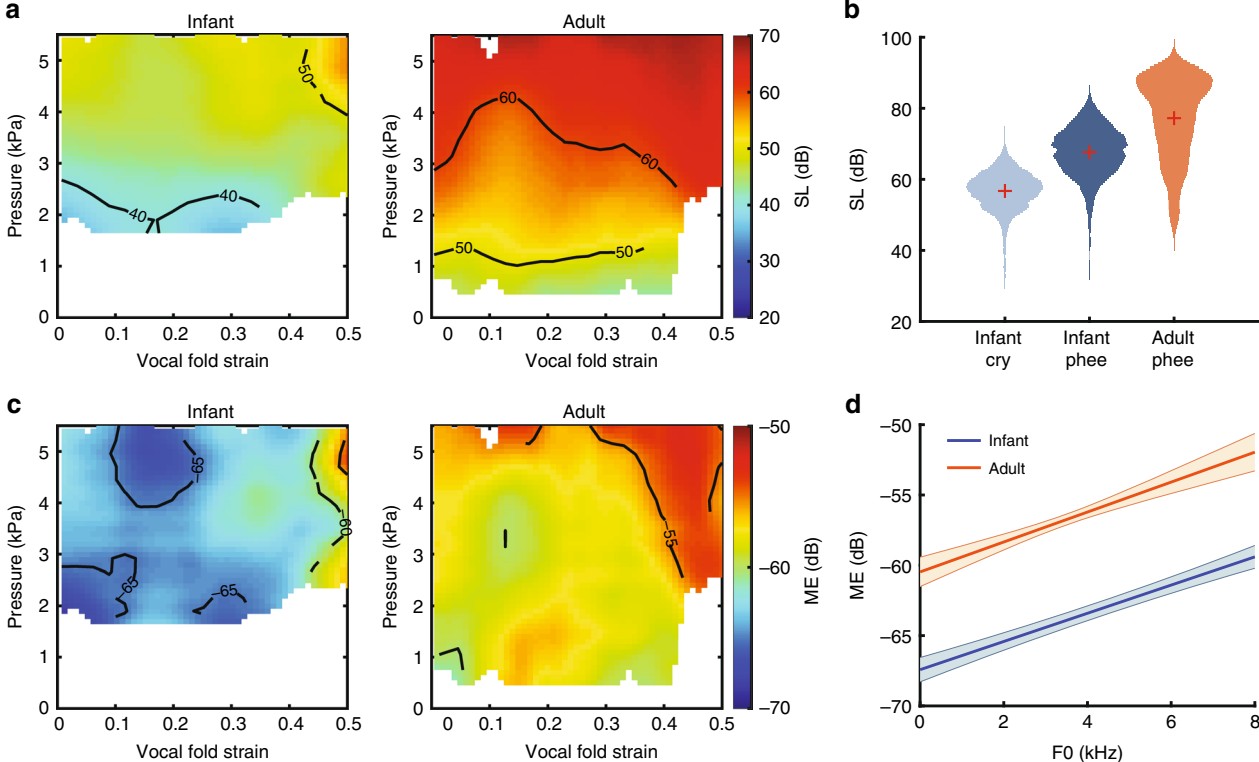

**Fig. 4** Adult larynx is more energy efficient than infant larynx. **a** Mean source level (SL) maps of infant ($n = 3$) and adult ($n = 4$) larynges. **b** Comparison of SL between marmoset infant cry ($n = 2524$ utterances from 10 infants), infant phee ($n = 3301$ utterances from 10 infants), and adult phee ($n = 433$ utterances from 9 adults). Red crosses indicate mean. **c** Mean mechanical efficiency (ME) of infant and adult larynges. **d** Linear regression of mechanical efficiency vs. $f_o$. Shaded areas indicate the 95% confidence interval. Source data are provided as a Source Data file.

calls when the VF strain lagged pressure at >90° (Fig. 5c). The in-phase control of pressure and VF strain that produces mature calls indicates synchronous activity of the laryngeal and respiratory muscles. To test this, we recorded electromyography signals from intramuscularly implanted electrodes in the cricothyroid (CT, i.e. the laryngeal muscle controlling VF length) and surface electrodes over the abdominal muscles (controlling subglottal pressure) in one adult marmoset. We observed that during mature calls, the CT and expiratory muscle activity was highly synchronized (Fig. 5d). The phase difference between expiration and CT muscle activity was near zero ($0.00 \pm 0.58$ rads, mean ± std, $n = 87$ calls; Fig. 5e), supporting the idea that the production of mature contact calls is due to strongly synchro-nized laryngeal (representing strain) and respiratory (represent-ing pressure) muscle activities.

Such phase-locked oscillations can emerge in the dynamics of coupled oscillators[42]. To show how the synchronized laryngeal–respiratory control may arise over the course of development (perhaps, facilitated by social reinforcement[41,43]), we generated a coupled oscillator model of laryngeal–respiratory CPGs, where their synchronization is positively correlated with their degree of coupling[42]. To account for the higher order control that may coordinate vocal CPG output[6], the model CPGs received a common input through which the CPGs are indirectly coupled (Fig. 5f). Within a range of parameter settings, the common input itself oscillates and forces the CPGs to oscillate at the same frequency with a fixed phase difference. The degree of coupling, negatively related to the phase difference between the CPGs, is controlled by the strength of this input (Fig. 5g). Based on this model, shifts in phase difference towards zero between pressure and strain over the first 2 months of life require increases

in the input strength from this putative higher order area to the CPGs (Fig. 5h, i).

## Discussion

Understanding the mechanisms underlying vocal behaviors remains a major challenge, and investigations of various species have revealed key neurobiological principles underlying vocal development but without considering the role of the changing body[44–46]. Across vertebrate taxa, changes occur during vocal development in a variety of parameters such as decreases[47] or increases[48,49] in fundamental frequencies, and increasing syllable repetition rates[49], but the contribution of the body—the vocal organ—remains unknown. Our work shows how gradual devel-opmental changes in the vocal organ itself can induce discrete state changes in vocal behavior. In infant marmoset monkeys, the transition between the two attractor states—from producing noisy, low fundamental frequency cries to tonal, higher pitched contact calls—occurs via a sound source switch from the VF proper to the VM. The adult larynx does not have this capacity. Because of its increased VM stiffness, the adult larynx only produces one attractor state characterized by stable, higher pit-ched sounds. Such changes in the material properties of the vocal organ, and its biomechanical consequences, can be used to locally 'compute' solutions, thereby significantly simplifying motor control and reducing the computational load, and thus energy consumption, of the developing brain[50]. Our work thus demon-strates the need for an embodied framework of vocal motor control[51], especially during postnatal development where the brain is navigating a rapidly changing body[19,20].

The mechanisms that account for the change in material properties of the developing marmoset larynx are unknown. In

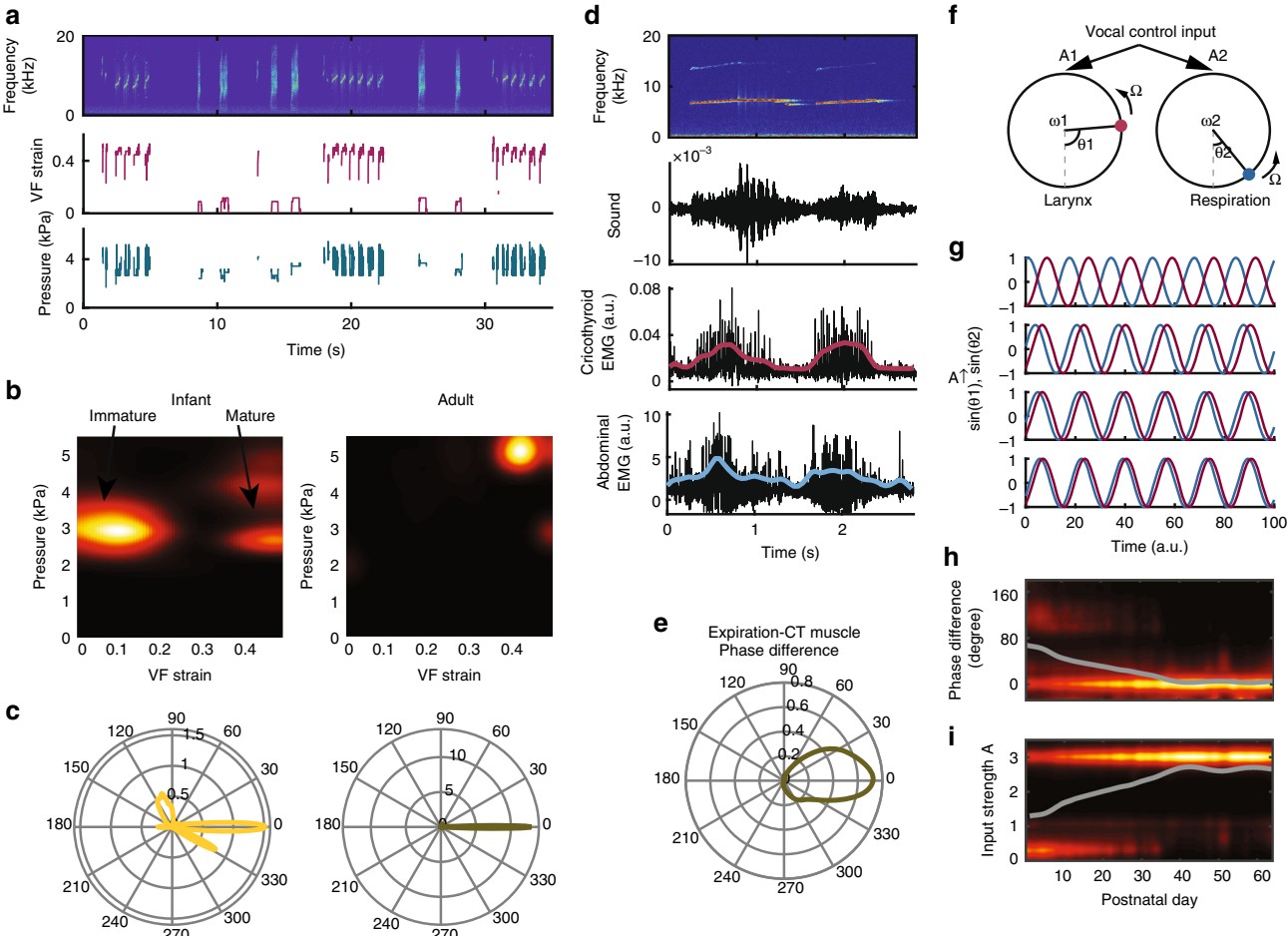

**Fig. 5** Vocal maturation requires the development of motor control. **a** An exemplar showing the reverse-engineering of vocal fold strain and subglottal pressure from a sequence of infant vocalizations. **b** The probability density of the infant and adult vocal control parameters in the space of subglottal pressure vs. vocal fold strain. A significant portion of the infant vocalizations were produced at the low-strain region, which is absent in the adult vocalizations. **c** The distributions of the phase difference between pressure and vocal fold strain during the production of infant and adult vocalizations. **d** EMG recordings at the right cricothyroid (CT) muscle and the abdominal muscle (for expiration) during adult phee call production. The amplitude modulation of the EMG signals is shown in bold (a.u., arbitrary units). **e** The distribution of the phase difference between the amplitude modulation of the expiratory muscle activity (corresponding to positive subglottal pressure) and the CT muscle activity. **f** A model for oscillators representing the respiratory and the laryngeal CPGs coupled with a common vocal control input. $A_{1,2}$ are the input strengths, $\omega_{1,2}$ are the intrinsic angular frequencies, $\theta_{1,2}$ are the phases of the two oscillators and $\Omega$ is the angular frequency of the input oscillator. **g** The phase difference decreases as the input strength $A$ increases in amplitude. **h** The probability density of the phase difference shifts towards zero over the first 2 months. **i** The probability density of the input strength $A$ increases over the first 2 months. The gray traces are the fitted trajectory.

most mammalian species studied, adult VFs commonly consist of multiple tissue layers, such as an epithelium, lamina propria, and muscle. The lamina propria itself is divided into several layers that differ in their composition of extracellular matrix proteins, such as glycosaminoglycans, and orientation and density of fibrous proteins, such as elastin and collagen[15]. Recently the embryological origins of tissues within the mammalian larynx were charted using genetic fate mapping[52], but much remains unknown regarding what molecular pathways effect postnatal protein expression in the larynx. Ultrastructural studies of the human VFs show that they change from a single layer at birth[13] to a multi-layered structure during infancy and puberty[15]. The lamina propria increases in thickness, and in its elastin content, from infant to adult, and continues to do so during aging[11]. Collagen in the VFs increases from infancy to adulthood as well[14], but remains stable thereafter[12]. Such material property changes should result in parallel changes in VF vibratory behavior and thus acoustics[24]. Indeed, it is known that mutations of the

elastin gene in humans lead to VF abnormalities and voice properties[53,54].

Our study focused on the contact call development—from cries to phees—of marmoset monkeys. This is because it is the only call in their repertoire that undergoes a significant transformation in acoustic properties during the course of development[17]. It's long duration, loudness and clear tonality (in mature versions) makes them uniquely difficult, more energetically costly, to produce when compared to other calls in the species repertoire[19]. Thus, while marmoset monkeys have a rich repertoire of vocalizations[55], all but the contact call sound adult-like at the beginning of postnatal life[17,56]. A number of call types, such as 'trills', 'trill-phees', and 'twitters', have a similar fundamental frequencies as the contact call[57,58]. However, these other call types are all very short in duration and without a sustained, tonally flat contour (that is initially noisy). Other call types in the species repertoire can sometimes have fundamental frequencies above 10 kHz ('tsik' calls) as well as below 1 kHz (like the 'egg' calls); these calls are

very short in duration[59]. Our study did not attempt to account for the production of these call types; there is no reason to think that the changing material properties of the larynx prevent the production of these other call types via a different set of laryngeal control dynamics. For example, the sub-1-kHz calls may be produced when the vocalis muscles within the VFs are shortened, a possibility that cannot be simulated in our experimental setup. Along similar lines, human VFs can exhibit several stable modes of vibration with different $f_o$ ranges, such as chest and falsetto regime[60]. We cannot exclude that multiple vibratory modes in VF or VM are possible as observed in the human larynx. For contact call development in marmosets, however, our data and neural modeling suggest that is the combination of changing laryngeal material properties (switching to VMs as the sound source) and the development of neural control of respiratory and laryngeal synchronization that transforms them from long-duration, noisy cries into long duration, tonal phee calls.

We provided the first experimental evidence for the role that VMs play in vocal production. Our finding that that they act as low-mass oscillators to produce high frequency vocalizations is consistent with previous hypotheses[29,31,47]; that these calls are louder and more efficiently produced in our study is consistent with model predictions for the role of VMs in mammals[30]. Although some coupling between VF and VM cannot be ruled out, our observations support earlier suggestions that apical VMs vibrate relatively independently from the VF to aid extremely high-frequency calls, allowing an even wider fundamental frequency range[29,32,47]. The VM likely also produces the high frequency vibration during echolocation and social calls in bats[61] and perhaps Felids, but this remains to be tested.

In conclusion, our study established that vocal state changes in infant marmoset monkeys are in part adaptive consequences of the changing laryngeal properties. The transition from low frequency cries to high-pitched phee calls is through a sound source switch from the VF to the apical VM, allowing the animal to produce louder vocalizations with higher efficiency. These results suggest that the computational load on motor control can be reduced by the optimal solutions provided by the peripheral biomechanics. For example, in addition to the material change to the larynx over the development, our neural model posits that the strength of an input from a higher order structure increasingly synchronizes laryngeal and respiratory CPGs resulting in the generation of mature contact calls. The strength of this input could be modulated by contingent social reinforcement by parents: infant marmosets who receive greater contingent responses from their parents produce fewer cries and more mature-sounding contact calls quicker[17,43], and infants who do not receive parental care continue to produce cries[40,41].

## Methods

**Subjects.** Excised larynges were collected opportunistically from deceased captive common marmosets (*Callithrix jacchus*) housed at Princeton University). The larynges were from three full-term newborns, one juvenile (male, 7 months old) and four adults (four males and one female, >2 years old). The larynx sound production experiments were conducted in three infants, one juvenile and four adult larynges. The stress/strain measurements were conducted for each side of the VFs from three infant and two adult larynges. The samples were collected at Princeton University, New Jersey, USA, stored in −80 °C and shipped to the University of Southern Denmark, Odense, Denmark, in dry ice. The in vivo recording of marmoset vocalization was from ten infants whose dataset was previously published (http://science.sciencemag.org/content/suppl/2015/08/13/349.6249. 523 734.DC1) and five adults, among which four were the same individuals used in the larynx experiments. One subject was used for the electromyographic recording at the laryngeal and respiratory muscles. All subjects used in this study were housed in the colony room maintained at ~27 °C temperature and 50–60% relative humidity with 12L:12D photoperiod. All subjects lived in family groups, had ad libitum access to water and were fed daily with a standard commercial diet supplemented with fruits and vegetables. We complied with all relevant ethical regulations for animal testing and research. All experiments were

approved by the Princeton University Institutional Animal Care and Use Committee (Protocol #1908–18).

**Experiment setup.** The larynx together with the trachea and the tongue were carefully extracted from the deceased animal. The day before the larynx experiment, the tissue stored in −80 °C was gradually warmed up in refrigerator overnight and was submerged in Ringers solution (5 °C)[62]. It was then transferred to a Sylgard-coated petri dish. The larynx was separated from the tongue, fat and extrinsic muscles. It was mounted with the rostral side up onto a polyethylene tube via the trachea and was secured using 5/0 silk suture. To expose the VF, the hyoid bone, the epiglottis, and the vestibular fold were removed. The larynx was surrounded with Ringers-solution-soaked Gelfoam to be kept humidified. To mark the position of the VF, two knots were made on the two ends of the VF membrane using 10/0 silk suture.

The tube coupled with the trachea was oriented vertically and was connected to the air supply system with bronchial pressure controlled by dual valve differential pressure PID controllers, with respect to atmospheric pressure. Airflow was measured with mass flow sensors (PMF series, Posifa Microsystems, San Jose, USA). The pressurized air supplied to the controllers was warmed up to 37 °C and humidified with deionized water in a pressure cooker. The tip of the ventral side of the thyroid cartilage was attached to the lever arm of the ergometer (Model 300C, Aurora Scientific, Ontario, Canada) via 5/0 suture threaded through a small hole made on the cartilage. The displacement of the lever was controlled to move in a negative sawtooth wave fashion, with the first 80% cycle stretching the VF and the last 20% cycle back to the zero-point at a rate of 2 Hz. To visualize the VF lengthening, a $5 \times 5 \times 5$ mm 45°-angled aluminum surface-coated prism (Thorlabs, Newton, New Jersey, USA) was placed on the right side of the larynx. The right side of the thyroid cartilage was removed to reveal the stereoscopic view of the VF in the prism. To cumulate enough subglottal pressure to power VF vibration, the arytenoid cartilages were adducted by two bent 27-gauge needles mounted on the micro-manipulators (model MM3, Narishige, Japan).

**Image acquisition and analysis.** To measure the dimension of the VF, a camera was mounted on a Leica M165-FC stereoscope (Leica Microsystems, Wetzlar, Germany). To record the varying VF length during phonation, a high-speed camera (MotionPro-X4, 12-bit CMOS sensor, Photron, Tokyo, Japan) was mounted on the microscope and the images were acquired at a rate of 300 Hz. In addition, the laryngeal dynamics was acquired with an ultra-high-speed camera (FASTCAM SA1, 16-bit, Photron, San Diego, CA, USA) at a frame rate of 30 kHz. Light for illumination was provided by a plasma light source (HPLS200, Thorlabs, Germany) through liquid light guides and reflected off a 45° prism to reduce heat. Video acquisition was automatically initiated by a synchronization signal upon the start of audio acquisition with a pre-fixed delay.

The momentary VF length was measured from videos taken from the 12-bit camera. The pixel distance for each video was calibrated. The suture marks were hand labeled in the images from the view of the prism at every 18 frames. The distance measurements corresponding to maximal stretches were also marked. As the stretching process was smooth, we resampled the sparsely measured VF length at 357 Hz using the cubic spline method with no smoothing.

To quantify the glottal areas enclosed by different depths of the VF, we used a supervised learning workflow implemented in ilastik (http://ilastik.org/)[63]. In short, we trained a random forest model to classify the cranial edges (the VM), the caudal edges (the VF) and the rest of the larynx. To calculate the enclosed areas bounded by VM and VF, we first skeletonized the detected pixels of the edges using MATLAB routine *bwmorph*, and then identified the boundaries of the edges using MATLAB routine *boundary*. For frames where the caudal edges were occluded by the cranial edges, we set the two areas equal.

**Data acquisition.** The audio signal was recorded with a ½-inch pressure microphone (46AD with pre-amplifier type 26AH, G.R.A.S., Denmark), amplified and high-pass filtered (above 10 Hz, 3-pole Butterworth filter, model 12AQ, G.R.A.S, Denmark). The microphone was calibrated before every experiment. The microphone was placed 7–10 cm away from the sample and the distance was measured each time. A customized MATLAB GUI was created to simultaneously record the acoustic signal, the bronchial pressure, the airflow, the ergometer displacement, the force (these are recorded at 40 kHz sample rate), as well as the internally triggered high-speed video data. The audio signal was corrected for the time delay of sound travel from the larynx to the microphone.

**Phonation threshold pressure.** To measure the PTP, we adducted the arytenoid cartilages and provided the larynx with ramping pressure while leaving the VF unstretched. The larynges were subject to ramping pressure up to 3.0 kPa at a rate of 1.0 kPa s$^{-1}$. PTP was identified as the pressure where the sound was first produced on the ramping up phase. A mild hysteresis was observed during the down phase. In this study, we only took the up phase into account.

**Control space construction for acoustic maps.** To systematically explore acoustic output in the control space of subglottal pressure and VF strain, we subjected the larynx to a set of subglottal pressure between 0 and 5 kPa with 0.5-kPa intervals

while manipulating the VF length from 0 to 50% strain. By slowly changing the VF length (sawtooth at 2 Hz), we predict that VF vibratory behavior was in a quasi-steady state. The strain of the VF was calculated as $\epsilon(t) = (L(t) - L_0)/L_0$, where $L_0$ is the VF length at rest. The maximal strain was set to ~50%. $f_o$ and WE were calculated on a sliding window of 12.8 ms with 10 ms overlap when sound was produced. The pressure and the VF strain corresponding to the midpoint of the sliding window were identified. To compare the acoustic range between infant and adult larynges, we calculated $f_o$ and WE within grids of VF strain (bin size = 0.05) and pressure (bin size = 0.5 kPa). If their vocal ranges were the same, one would expect that the acoustic features within the same grid were the same and they should fall onto the 45° line of the infant vs. adult plots. We thus compared the slopes of those plots to 1. A slope >1 means a greater range in the infant larynx than the adult's, and vice versa.

**Elastic property measurement.** The larynges were cut in half along the midline, and the VFs were dissected with a small portion of the arytenoid and thyroid cartilages attached on the two ends. The membrane was isolated from the VF. The specimen was mounted onto the two ergometer arms of a uniaxial force transducer (Model 400A, Aurora Scientific) using silk suture along the dorsal-ventral direction. The specimen was submerged in Ringer's solution at a physiological pH of 7.4 and a temperature of 38 °C, resembling the marmoset body temperature. The elastic properties of the specimens were quantified by real-time measurement of force and displacement. The samples were preconditioned with 0.1 mm sinusoidal strain waves at 1 Hz of three cycles each starting position and then were subjected to 20 cycles of a sinusoidal displacement signal of 50% strain at 1 Hz. The force and displacement signals were filtered with third-order low-pass Butterworth filter with a cutoff frequency at 100 Hz.

The strain was defined similarly as above:

$$\epsilon = \frac{\Delta L}{L_0}, \tag{1}$$

where $L_0$ is the resting length and $\Delta L$ the displacement. The stress was defined as:

$$\sigma = \frac{F}{A}, \tag{2}$$

where $F$ is the force measured by the stretch apparatus and $A$ the cross-sectional area of the specimen. Usually the volume of the VF is considered incompressible[34,64–66], the cross-sectional area must shrink as the tissue is stretched. The area as a function of strain can be estimated as:

$$A = \frac{V}{L} = \frac{m}{\rho L_0 (1 + \epsilon)}, \tag{3}$$

where $m$ is the VF (or membrane) mass and $\rho = 1.04 \times 10^3$ kg m$^{-3}$ the commonly used density in VF studies.

The stress–strain response was modeled with a linear model at low strain and an exponential model at high strain:

$$\sigma = \begin{cases} a + b\epsilon, & \epsilon \leq \epsilon_0 \\ Ae^{B\epsilon}, & \epsilon > \epsilon_0 \end{cases}, \tag{4}$$

where $\epsilon_0$ is the linear limit. To satisfy the continuous and differentiable requirements, we set $a + b\epsilon_0 = Ae^{B\epsilon_0}$ and $b = ABe^{B\epsilon_0}$, and thus the linear limit must have $\epsilon_0 = \frac{1}{B} - \frac{a}{b}$. The parameters were determined by fitting the model using MATLAB routine *lsqcurvefit*.

To plot acoustic features in the pressure–stress space, we converted VF strain into stress using the fitted model for the VM stress–strain response.

**Efficiency analysis.** To estimate the SL of the emitted sound from the excised larynx, we calculated the root mean square (RMS) of the calibrated sound pressure within 12.8 ms sliding windows. The SL at 1-m distance was estimated using

$$SL = 20\log_{10}\frac{p}{p_0} + 20\log_{10}r, \tag{5}$$

where $p$ is the RMS of sound pressure, $p_0 = 20$ μPa is the reference pressure, and $r$ the distance from the excised larynx to the microphone.

To calibrate the SL of marmoset vocalizations, a speaker broadcasting a constant 45-dB SPL pink noise was placed in the room to mask occasional noises. The sound pressure of marmoset vocalization was calibrated to the background noise. A microphone was positioned at a distance of 0.76 m to the testing box where the marmoset was housed. The RMS of the calibrated sound pressure was used to calculate the SL using the above formula.

The ME was calculated using

$$ME = 10\log_{10}\frac{P_{acoustic}}{P_{aerodynamic}}, \tag{6}$$

where the acoustic power in Watts $P_{acoustic} = IA = \frac{4\pi r^2 p^2}{\rho v}$, and the aerodynamic power in Watts $P_{aerodynamic} = P_s \dot{V}$. Here we used air density $\rho = 1.2$ kg m$^{-3}$, speed of sound $v = 344$ m s$^{-1}$, $p$ the RMS sound pressure in Pa, $P_s$ the subglottal pressure in Pa, and $\dot{V}$ the glottal flow rate in m$^3$ s$^{-1}$ (refs. [39,67]).

**Statistical analysis.** To compare the stress–strain response between the VF and the VM, as well as the difference in the response due to age, we used a multiple linear regression model:

$$\ln \sigma = a + b * \epsilon + c * type + d * age + e * \epsilon * type + f * \epsilon * age + g * type * age + h * \epsilon * type * age + error, \tag{7}$$

where type (VF or VM) and age (adult or infant) are dummy variables, which were coded as 0 or 1 for each category.

To estimate the strain where the VM exceeded the VF in stress, we randomly resampled the stress-strain curves for each age and specimen type group with replacement for 1000 times and calculated the difference between the mean of the VM and VF curves for each repetition. The divergent point was determined as the point where the lower bound of the 95% confidence interval of the difference crossed zero. To estimate the mean and the uncertainty of this point, we repeated the bootstrap step for 1000 times. The $p$-value corresponding to the difference in the mean divergent point between infant and adult was estimated by assuming normality of the distributions.

To compare the SL among infant cry, infant phee call, and adult phee call, ANOVA was performed in combination with Tucky's honest significant difference post hoc test.

To test the SL (or ME) change in response to pressure and VF strain, we carried out multiple linear regression:

$$SL = a + b\epsilon + cP_s + error, \tag{8}$$

where $\epsilon$ is the VF strain and $P_s$ the subglottal pressure, for infant and adult larynges, respectively.

**Control parameter reconstruction.** The control parameters were estimated on a sliding window of 12.8 ms with 10 ms overlap. Within each window, $f_o$ and WE were calculated and then z-score-normalized. In the normalized $f_o$–WE space, a tolerance circle of radius 1 (1-sigma away from the center) was created. Within this circle, the control parameters closest to the previous action in Euclidean distance were identified as the control parameters generating the concurrent sound. The control parameters were only evaluated during vocalization, and the initial conditions for VF strain and pressure were set to 0 and the PTP, respectively. The 2D probability density of the control parameters was constructed from all the momentary control parameters.

To construct the polar density, the VF lengthening and pressure were converted to values between $-\pi/2$ and $\pi/2$ corresponding to no lengthening (or non-positive pressure) and maximum lengthening and pressure. The polar density was calculated using MATLAB exchange file *circ_ksdensity*.

**Intramuscular electromyography (EMG) surgical procedure.** All procedures were approved by the Princeton University Institutional Animal Care and Use Committee. The implant surgery was performed under sterile conditions with the animal deeply anesthesia with isoflurane (1.0–2.0%) in oxygen gas. Vital signs including heart rate, respiratory rate, SPO$_2$, and body temperature were monitored throughout the surgery. A small incision was made to expose the skull for connector attachment. A connector A75859) soldered with stainless steel wire electrodes (A-M Systems 790900) was secured on the surface of the skull with adhesive cement (C&B Metabond). Each pair of electrodes was attached to a 27-gauge needle tip with polyimide tubing protected. The animal was then rotated to expose the ventral side of the neck. A 2-cm-long incision between the pogonion and the cranial end of the sternum was made on the midline ventral neck. To expose the cricothyroid muscles, subcutaneous fat, glandula mandibularis and sternohyoideus muscles were separated using blunt scissors. All electrode wires were passed from the skull subcutaneously. Around 3 cm from the tip on each pair of electrodes, a 1-mm opening was made on the insulation (as electrode contacts) with 2-mm separation between the two wires. A bead made from colored Kwiksil was placed next to the contact for visualization. The needle was passed twice through the fascia on top of the CT muscle using a needle driver till the beads touched the fascia and a suture knot was made to secure the wires in place. The needle tip and excessive wire were removed. After the electrodes implanted, the incisions were closed using 5/0 suture with a drop of Vetbond applied.

**EMG signal acquisition and processing.** The EMG signal of CT muscle activity was recorded from the intramuscular implant and that of the abdominal muscle activity was recorded from a pair of Ag-AgCl surface electrodes (Grass technology) attached to the torso of the subject with an elastic band. The EMG signal from each pair of electrodes was differentially amplified (250×) and the resulting signal was acquired by Plexon Omniplex System. The signal was notch-filtered at 60 Hz and digitized at a sample rate of 40 kHz.

The EMG signal was bandpass filtered between 100 Hz and 3 kHz using a zero-phase elliptic filter. The signal was rectified. The amplitude modulation of the EMG signal was calculated as the envelope of the rectified signal. Vocalizations during EMG recording were identified using threshold crossing. The EMG signal 0.5 s before and 0.5 s after vocal production was extracted.

To calculate the phase difference between abdominal and CT muscle contraction, we calculated the cross-correlation between the amplitude modulation of the two EMG signals around phee calls. The time delay $\tau$ was determined as the location of the first peak of the cross-correlation. We then calculated the mean duration $d$ of the utterances. The phase difference was then estimated as $\delta\phi = 2\pi\tau/d$.

**Coupled oscillator model.** We simulated the oscillatory dynamics for the CPGs of the laryngeal and respiratory activities using circular oscillators coupled with a common oscillatory input. The equations are given as follows:

$$\begin{cases} \dot{\theta}_1 = \omega_1 + A_1\sin(\phi - \theta_1) \\ \dot{\theta}_2 = \omega_2 + A_2\sin(\phi - \theta_2), \\ \dot{\phi} = \Omega \end{cases} \qquad (9)$$

where $\theta_{1,2}$ are the respiratory and the laryngeal phases, $\omega_{1,2}$ ($\omega_1 > \omega_2$) the intrinsic angular frequencies, $A_{1,2}$ the input strength, $\phi$ and $\Omega$ the input phase and frequency, respectively. For illustration purpose, we let $A_1 = A_2 = A$ and $\Omega = \frac{\omega_1 + \omega_2}{2}$ (these conditions are not necessary). When $\Omega$ fixed, synchronization between the two oscillators occurs when $A > \frac{(\omega_1 - \omega_2)}{2}$. The phase difference between the two oscillators is thus $= \mu_1\sqrt{1 - \mu_1^2} - \mu_2\sqrt{1 - \mu_2^2}$, where $\mu_i = \frac{\Omega - \omega_i}{A}$, $i = 1, 2$. Obviously, $\theta_1$ and $\theta_2$ tend to oscillate in phase as the input strength $A$ increases.

To estimate the control input strength for marmoset development data, we first calculated the phase difference between pressure and VF length for the vocalizations produced during the first 2 months. The values were then used to estimate $A$. Here we let $\omega_1 = 1$, $\omega_2 = 0.5$, and set the boundaries for $A$ between 0.25 and 3.

**Reporting summary.** Further information on research design is available in the Nature Research Reporting Summary linked to this article.

## Data availability
Data are provided for Figs. 1c, d, 2c, h, i and 4b in Source Data file. The remaining datasets and codes that support the findings of the current study are available in the DRYAD Digital repository (https://doi.org/10.5061/dryad.h5r7m66).

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

## Acknowledgements

We like to thank Lasse Jakobsen for equipment loan. This research was supported by the NIH NINDS (R01NS054898) to A.A.G. and the Danish Research Foundation and NovoNordisk Foundation (NNF17OC0028928) to C.P.H.E.

## Author contributions

Y.S.Z., A.A.G., and C.P.H.E. conceived the study and designed the experiments. Y.S.Z., D.Y.T., D.A.L., and C.P.H.E. conducted the research. Y.S.Z. and C.P.H.E. performed data analysis. Y.S.Z., A.A.G., and C.P.H.E. wrote the initial draft. Y.S.Z., A.A.G., and C.P.H.E. edited and reviewed the final manuscript.

## Competing interests

The authors declare no competing interests.
