## [Peer Review File · Nature Communications]

Reviewers' Comments:

Reviewer #1:

Remarks to the Author:

In the present work, Zhang and colleagues investigated the role of biomechanical properties of the marmoset larynx on vocal motor production during postnatal development – a largely overseen topic. Using a combination of a number of well-controlled, elaborated methodological approaches in a sufficient number of animals/larynges they convincingly show that the transition from immature infant cries to adult phee calls seems to be mainly caused by the developing larynx and not the underlying brain circuits. This transition is mainly due to a sound source switch from vocal fold to vocal membrane oscillations.

This is an exciting paper and important work on the largely underestimated role of the developing vocal apparatus during vocal development. The results are novel and compelling and the manuscript is well-written. This work will be of broad interest and suitable for publication in Nature Communications once my concerns below have been sufficiently addressed.

Major comments:

1. Based on their experimental results, the authors explain the loss of cry vocalizations after infancy by a change in the biomechanical properties of the marmoset larynx and hypothesize that “the cry attractor state is outside the vocal capacity of the adult larynx”. However, recent studies clearly show that marmoset monkeys, which experienced limited parental feedback during infancy and adolescence do still produce cry vocalizations in certain behavioral contexts at adult age (Gultekin and Hage, Nat Commun 2017, Gultekin and Hage, Sci Adv, 2018). However, the vocal performance of these animals should not be existent with respect to the model proposed in the present study. It will be important to explain how these animals fit into the proposed framework of the present study.

Minor comments:

1. In line 237 the authors state that the larynges were collected from deceased monkeys, in line 256 they write that the larynges were extracted from euthanized monkeys. Please clarify.
2. Figure 2: The comparison of WE between infant and adult larynges generated with the same pressure and vocal fold strain settings show slightly lower yet high WE values also for the adult larynges (above -20dB). More specifically, it seems that adult larynges are also able to produce sounds with WE values well above -20dB and therefore within the WE range of infant cries (see Fig 1c). Please clarify.

Reviewer #2:

Remarks to the Author:

This paper provides evidence for the hypothesis that vocal development in marmoset monkeys is guided largely by biomechanical changes in the vocal folds, not only by brain development. It is shown that the infant marmoset has a capacity for self-sustained oscillation in two modalities, while the adult vocalizes in one modality. One modality is low-frequency, low periodicity oscillation of the vocal folds proper, which the infant uses in its cries. The other modality is high frequency, high periodicity oscillation of a vocal membrane that extends from the superior-medial edge of the vocal folds, which the adult uses in its mature “phee” vocalizations. Multiple methodologies are used, including high-speed imaging of vocal fold motions in excised larynges, stress-strain analysis of vocal fold tissues,

acoustic analysis, and phonation threshold (PTP) measurement.

While the results add some validity to the hypothesis, they do not propose anything fundamentally new. In most fields of motor learning, it has been recognized that the brain develops in conjunction with, and often as a result of, changes in the physical plant. In humans, for example, infant cries are rough and aperiodic because the alignment of collagen and muscle fibers is slow to develop. The full development of the vocal ligament may take several years. Laryngeal musculature is activated to enhance this development, and proper control then requires new muscular strategies

The methodology is weakened by the fact that no vocal tract is present when excised larynges are used to study self-sustained oscillation. The video images shown here do not clearly reveal whether or not there is an alternation of convergent-divergent glottal shapes (upper-lower phase difference) when the vocal membrane vibrates. This would appear to be a requirement without vocal tract interaction. [The references cited include mathematical treatments of how an acoustically inertive vocal tract can interact with a sound source to produce self-sustained oscillation when there is no upper-lower phase difference]. The conclusion drawn by this reviewer is that (1) membrane development basically follows ligament development in humans, (2) it is likely that the membrane is not vibrating in isolation, but some coupled movement exist between the membrane and the lower fold, and (3) the two vibration modalities are similar to chest and falsetto registration found in many species.

Reviewer #3:

Remarks to the Author:

In this study, the authors explore the relationship between vocal production and the changing physical and material properties of the vocal tract. Combining theory and experiment, the authors present data supporting the hypothesis that changing laryngeal dynamics need to be taken into account in order to explain vocal development, i.e., vocal development is not a neural control signal change alone. This is a fascinating study that builds upon the earlier work of some authors, and will be of great interest to a broad range of neuroscientists. However, the paper is not written to be broadly accessible. Control experiments are not discussed or described, and a key prediction of the model (phase differences) is untested. The discussion, for a topic of such wide interest, is frustratingly brief to non-existent. Therefore, while I am positive about this very interesting manuscript, I would like to see the authors address some concerns in a major revision.

Major concerns

1) My biggest concern is essentially the lack of a control experiment. While the maturation and material changes of vocal tract properties are explored in this study for the cries → phee transition, nothing is said about the other sounds that are produced using the same apparatus, yet do not appear to acoustically change over development (twitters or trills, as discussed in Takahashi et al 2015). These sounds also appear to be of high-frequency like the phee, so similar issues would hold. On the flip side, it does not seem to be the case that adults are incapable of producing low-frequency sounds – for example, Agamaite et al 2015 describe “Egg” and “Ock” sounds that have sub-1KHz f0s. If it were the case that the control space for low-frequency production is lost in adults, how are they still able to produce these sounds?

2) Which control parameter is responsible for the loudness of the produced sounds? The contact calls studied here seem extremely loud, and it appears that if one emphasizes loudness as a critical factor, you could take an underlying linear system and over-drive it to produce nonlinear outputs with

harmonic distortions. Could this be an alternative model to explain why a developmental change is seen for phee but not other, quieter, sounds?

3) Related to the authors' earlier study, how do the authors imagine an instructive signal (such as parental feedback) facilitates the transformation from cries → phee? Is it entirely based on the synchronization of respiration and muscle tension as their model suggests? But then wouldn't marmosets also need to learn synchronization for the other sounds they produce?

4) In Fig. 2, it seems that the experiments were performed by holding one parameter and ramping the other parameter. Given the modeling in Fig. 4, it would be far more interesting to co-vary the parameters in-phase or out-of-phase. In an adult vocal tract, maybe it is possible to generate low frequency sounds after all by changing this phase difference. This would also go towards addressing concern (1) above.

5) Please state the mean lengths of the vocal folds in infants and adults. VF length and VF strain are used interchangeably when describing Fig. 2 results, while these are of course related, it would help to consistently stick to absolute or relative units. It would be instructive to also provide, as extended data figure, the plots in 2e-g, with VF *stress* on the x-axis, as the authors mention in line 125 that this could be a critical factor in generating sounds with high f0s.

6) Why are there systematic differences in the sampling of the pressure-strain space between adults and infants? Looks like the minimum pressure applied was ~2kPa for infants and ~0.5 kPa for adults. The reason for this is not clear from the Methods. What do the blank regions in these plots signify – data not acquired, or f0 not quantifiable? Since the f0 of the infant cry is ~500Hz, it would be useful to mark this contour in Fig. 2e to g.

7) Are these systematic differences in the VF thickness between infants and adults that could explain some of the differences in the stress-strain curves in Fig. 3b for VF? On a related note, I would suggest plotting VM and VF on separate plots, it is difficult to compare across adult and infant VF and VM, partly because it is scaled for VM. To my eye it seems that the infant stresses at high strains are ~50% larger than adults, and this could contribute to the steeper slopes seen in Fig. 2 (another reason to show pressure-stress axes).

8) The paragraph beginning line 121 could be rewritten much more clearly. If a string model were true, adults would be at an advantage for producing lower frequency sounds (longer VF, smaller non-linearity in stress-strain curves). Yet they don't, suggesting that the string models are missing an important component. Then introduce the VM. A better picture of the larynx showing the VF and VM would be helpful.

9) The manuscript is highly technical and hard to read. It is not written with a general audience, such as the readership of Nature Communications, in mind. The authors seem to be well within the available word limit, and I would highly recommend that they provide more narrative explanations and interpretations. The explanation for Fig. 4 is particularly dense and hard to follow.

Additional Comments

1) Please explain you term the peaks of the parameter distributions in Fig. 1c - f "attractors". This terminology is introduced in the results section without

2) In Fig. 1 it would be helpful to include the spectra of the cry, subharmonic, and phee with f0 marked as in Ext. Fig. 1b.

3) The red and blue colors for the lines in Fig. 2d should be cued better and defined in the legend. Maintain the same color scheme in Ext. Fig. 2 (it is currently flipped).

4) In the supplementary movies, could you mark the orientation of the cranial and caudal sides? The difference referred to in line 163 is not at all clear in Movie S1.

5) Line 179 The authors state that the adult VM has increased stiffness, but from Fig. 3b it seems that the adult VM slope is actually smaller than the infant VM slope. Is this a typo?

6) line 193 For infants, the two *attractors *

Reviewer #1 (Remarks to the Author):

In the present work, Zhang and colleagues investigated the role of biomechanical properties of the marmoset larynx on vocal motor production during postnatal development – a largely overseen topic. Using a combination of a number of well controlled, elaborated methodological approaches in a sufficient number of animals/larynges they convincingly show that the transition from immature infant cries to adult phee calls seems to be mainly caused by the developing larynx and not the underlying brain circuits. This transition is mainly due to a sound source switch from vocal fold to vocal membrane oscillations.

This is an exciting paper and important work on the largely underestimated role of the developing vocal apparatus during vocal development. The results are novel and compelling and the manuscript is well-written. This work will be of broad interest and suitable for publication in Nature Communications once my concerns below have been sufficiently addressed.

We thank the reviewer for his/her time and effort to review our paper, encouraging words and constructive comments. In the revised manuscript we have addressed the concerns that he/she has raised, as described below.

Major comments:

Q1. Based on their experimental results, the authors explain the loss of cry vocalizations after infancy by a change in the biomechanical properties of the marmoset larynx and hypothesize that “the cry attractor state is outside the vocal capacity of the adult larynx”. However, recent studies clearly show that marmoset monkeys, which experienced limited parental feedback during infancy and adolescence do still produce cry vocalizations in certain behavioral contexts at adult age (Gultekin and Hage, Nat Commun 2017, Gultekin and Hage, Sci Adv, 2018). However, the vocal performance of these animals should not be existent with respect to the model proposed in the present study. It will be important to explain how these animals fit into the proposed framework of the present study.

A1. The reviewer makes an excellent, insightful point. We did not mean to imply that vocal change is solely due to the changing material properties of the larynx; this is the reason why we included the neural model. The reviewer’s insight gave us an opportunity to better motivate this model. In the revised results, we now begin the neural model section with the following:

“Our findings do not address the role of the concurrently changing nervous system—the vocal CPGs^{18,19} and descending control³. In the case of marmoset monkeys, this is important because infants raised without parental care continue to produce noisy, cry-like vocalizations well beyond the age when such calls should have disappeared from the vocal repertoire^{48,49}. These animals are not stunted in their bodily growth and it is unlikely that their laryngeal development is dependent upon parental care. It is therefore logical to infer a role for experience-dependent neural control.”

Minor comments:

Q2. In line 237 the authors state that the larynges were collected from deceased monkeys, in line 256 they write that the larynges were extracted from euthanized monkeys. Please clarify.

A2: We changed “euthanized” to “deceased”.

Q3. Figure 2: The comparison of WE between infant and adult larynges generated with the same pressure and vocal fold strain settings show slightly lower yet high WE values also for the adult larynges (above -20dB). More specifically, it seems that adult larynges are also able to produce sounds with WE values well above -20dB and therefore within the WE range of infant cries (see Fig 1c). Please clarify.

A3: Thank you for the opportunity to clarify. Two factors contribute to WE value above -20 dB in adults:

1. The points between -20 and -10 dB in Fig 2I represent combinations of high pressure (>5 kPa) and low strain (top left in figure 1G panel Adult). In this regime, the produced sound did not have a clear f_0 , and seemed chaotic and presumably caused by irregular VM oscillations. However, the sounds produced by infant larynx at low strain and pressure <5 kPa were periodic with low f_0 , similar to the acoustic structure of infant cry. Thus, the high WE, chaotic sound corresponding to the top left corner of the adult map is different from the infant cry-like sound. The WE cannot distinguish dense harmonic stack from noise. In figure 5, we argue that the adult phee is made at 5 kPa and .4 strain, so this chaotic regime at low strain and high pressure in vitro is likely not used by the adults in vivo.

2. Overall the WE values of adults were slightly higher because we did not add a vocal tract to our in vitro recordings. Comparing Fig 1a and 2D; the harmonics are much stronger in vitro because there is no vocal tract acting as a band-pass filter. We calculated WE in the frequency range up to 20 kHz, so more harmonics are included for low f_0 sound than high f_0 sound within the same spectral range. Thus low f_0 is naturally correlated with WE because of the harmonics.

Reviewer #2 (Remarks to the Author):

This paper provides evidence for the hypothesis that vocal development in marmoset monkeys is guided largely by biomechanical changes in the vocal folds, not only by brain development. It is shown that the infant marmoset has a capacity for self-sustained oscillation in two modalities, while the adult vocalizes in one modality.

One modality is low frequency, low periodicity oscillation of the vocal folds proper, which the infant uses in its cries. The other modality is high frequency, high periodicity oscillation of a vocal membrane that extends from the superior-medial edge of the vocal folds, which the adult uses in its mature “phee” vocalizations. Multiple methodologies are used, including high-speed imaging of vocal fold motions in excised larynges, stress-strain analysis of vocal fold tissues, acoustic analysis, and phonation threshold (PTP) measurement.

Q1: While the results add some validity to the hypothesis, they do not propose anything fundamentally new.

A1: We thank the reviewer for his/her time and effort to review our paper. In the revised manuscript we have addressed the concerns that he/she has raised, as described below.

The marmoset monkey is an increasingly popular model for human behavior in the neurosciences (Belmont et al., *Neuron*, 2015; Miller et al., *Neuron*, 2016). Specifically, they are a well-suited model to study primate vocal communication system because their vocal development shares many features with prelinguistic human vocal development. As we write in our Introduction, the transition from noisy to tonal vocalizations is something humans and marmosets share. The mechanisms behind this “cries to phee” transition in marmosets have been the subject of recent debate in a series of papers (Takahashi et al 2015, 2017; Teramoto et al. 2017; Gultekin and Hage 2017, 2018; all cited in the manuscript). These papers suggest that neural drive is the major factor driving the transition over development.

In our paper, we experimentally test the alternative hypothesis that previously unappreciated contributions of functional changes in the larynx can explain the observed vocal state changes during development. We agree with the reviewer that the methodology of studying high-speed imaging of vocal fold motions in excised larynges, stress-strain analysis of vocal fold tissues, acoustic analysis, and PTP measurements each for themselves are not fundamentally new methods. However to the best of our knowledge, we introduce the first motorized excised larynx experiments to systematically explore the vocal control space in any vertebrate, and we do so in the context of development. We furthermore propose a novel empirically based computational model explaining how established descending motor control drives laryngeal dynamics, leading to normal vocal development.

We used this combination of novel and existing methodologies to generate experimental datasets that are completely new and unpublished for marmosets.

In humans, a large body of literature exists on laryngeal anatomy and vocal fold microanatomy, including e.g. collagen distribution, studied at different ages from newborns to the elderly (seminal and inspiring work by Titze, Hirano, Sato, Hammond and others cited in our manuscript). However we are not aware of any microanatomy papers in marmosets or of longitudinal studies reporting tensile tissue properties in humans over development, only

adults. Thus our hypothesis that the dynamics of infant vocalizations are related to functional changes of the developing larynx is difficult to test experimentally in humans.

In order to better contextualize our work in the existing literature, we added a paragraph to the Discussion. This also accommodates the reviewer's suggestion in Q5.

“The mechanisms that account for the change in material properties of the developing marmoset larynx are unknown. In most mammalian species studied, adult vocal folds commonly consist of multiple tissue layers, such as an epithelium, lamina propria and muscle. The lamina propria itself is divided into several layers that differ in their composition of extracellular matrix proteins, such as glycosaminoglycans, and orientation and density of fibrous proteins, such as elastin and collagen²⁷. Recently the embryological origins of tissues within the mammalian larynx were charted using genetic fate mapping⁵⁷, but much remains unknown regarding what molecular pathways effect postnatal protein expression in the larynx. Ultrastructural studies of the human vocal folds show that they change from a single layer at birth²⁵ to a multi-layered structure during infancy and puberty²⁷. The lamina propria increases in thickness, and in its elastin content, from infant to adult, and continues to do so during aging²³. Collagen in the vocal folds increases from infancy to adulthood as well²⁶, but remains stable thereafter²⁴. Such material property changes should result in parallel changes in VF vibratory behavior and thus acoustics³³. Indeed, it is known that mutations of the elastin gene in humans lead to vocal fold abnormalities and voice properties^{58,59}.”

Q2: In most fields of motor learning, it has been recognized that the brain develops in conjunction with, and often as a result of, changes in the physical plant.

A2: We wholeheartedly agree with the reviewer that the brain develops in conjunction with, and often as a result of, changes in the physical plant. In our experience this is however still not a mainstream idea in many fields of motor learning, where the focus is on neural aspects such as circuitry formation and plasticity, and not behavior (e.g. Krakauer et al 2017) or embodiment (e.g. Pfeiffer and Bongard, 2007).

As mentioned above specifically the phee-cry transition in marmosets has been the subject of recent debate study in a series of papers. These papers allude to neural drive being a major factor. Here we test and show for the first time that bodily changes can explain in part the observed behavioral changes over development. Thus, this work further advocates the important view that changes in the physical plant are essential to incorporate when studying vocal development in particular, and motor development in general.

Q3: In humans, for example, infant cries are rough and aperiodic because the alignment of collagen and muscle fibers is slow to develop. The full development of the vocal ligament may take several years. Laryngeal musculature is activated to enhance this development, and proper control then requires new muscular strategies

A3: In humans, it has been established that anatomical vocal fold maturation is not complete until about 15 years of age in boys (Titze in Vocal fold physiology, cited in our manuscript). Interestingly the human larynx is already fully capable of intelligible speech and song by this age. We are unaware of studies that compute control strategies over development in humans, probably because we cannot drive physiological changes in human vocal folds and have very limited success in measuring single unit EMG of laryngeal muscles. Given the dedicated neural, muscular and genetic substrates, the knowledge accrued to date, and the increasing number of genetic tools (from RNA-interference to DREADDs and optogenetics) available to

the field, comparative model systems such as marmosets and songbirds are ideal systems to embrace an integrative approach to casually test these correlations. This discussion is outside the scope of this paper.

Q4: The methodology is weakened by the fact that no vocal tract is present when excised larynges are used to study self-sustained oscillation. The video images shown here do not clearly reveal whether or not there is an alternation of convergent-divergent glottal shapes (upper-lower phase difference) when the vocal membrane vibrates. This would appear to be a requirement without vocal tract interaction. [The references cited include mathematical treatments of how an acoustically inertive vocal tract can interact with a sound source to produce self-sustained oscillation when there is no upper-lower phase difference].

A4: A vocal tract is essential for a single mass model to exhibit self-sustained oscillation. The marmoset VMs clearly exhibit self-sustained oscillations without a tract present, so some energy transfer from fluid to tissue must be present most likely as a caudocranial wave. This energy conversion is not the main point of this paper and does not affect our main conclusions.

Importantly, the role of the vocal tract in the vocalizations of developing marmosets was recently tested. The data suggest that it has only a very small influence on their fundamental frequencies (Zhang and Ghazanfar, 2018).

Q5: The conclusion drawn by this reviewer is that (1) membrane development basically follows ligament development in humans, (2) it is likely that the membrane is not vibrating in isolation, but some coupled movement exist between the membrane and the lower fold, and (3) the two vibration modalities are similar to chest and falsetto registration found in many species.

A5: To accommodate these comments we have added the following two paragraphs in our revision:

“We provided the first experimental evidence for the role that vocal membranes play in vocal production. Our finding that that they act as low-mass oscillators to produce high frequency vocalizations is consistent with previous hypotheses^{38,39,52}; that these calls are louder and more efficiently produced in our study is consistent with model predictions for the role of vocal membranes in mammals¹. Although some coupling between VF and VM cannot be ruled out, our observations support earlier suggestions that apical VMs vibrate relatively independently from the VF to aid extremely high-frequency calls, allowing an even wider fundamental frequency range^{38,40,52}. The VM likely also produces the high frequency vibration during echolocation and social calls in bats⁶⁶ and perhaps Felids, but this remains to be tested.”

“The mechanisms that account for the change in material properties of the developing marmoset larynx are unknown. In most mammalian species studied, adult vocal folds commonly consist of multiple tissue layers, such as an epithelium, lamina propria and muscle. The lamina propria itself is divided into several layers that differ in their composition of extracellular matrix proteins, such as glycosaminoglycans, and orientation and density of fibrous proteins, such as elastin and collagen²⁷. Recently the embryological origins of tissues within the mammalian larynx were charted using genetic fate mapping⁵⁷, but much remains unknown regarding what molecular pathways effect postnatal protein expression in the larynx. Ultrastructural studies of the human vocal folds show that they change from a single layer at birth²⁵ to a multi-layered structure during infancy and puberty²⁷. The lamina propria increases in thickness, and in its elastin content, from infant to adult, and continues to do so during aging²³. Collagen in the vocal folds increases from infancy to adulthood as well²⁶, but remains stable thereafter²⁴. Such material property changes should result in parallel changes in VF vibratory behavior and thus acoustics³³. Indeed, it is known that mutations of the elastin gene in humans lead to vocal fold abnormalities and voice properties^{58,59}.”

Reviewer #3 (Remarks to the Author):

In this study, the authors explore the relationship between vocal production and the changing physical and material properties of the vocal tract. Combining theory and experiment, the authors present data supporting the hypothesis that changing laryngeal dynamics need to be taken into account in order to explain vocal development, i.e., vocal development is not a neural control signal change alone. This is a fascinating study that builds upon the earlier work of some authors, and will be of great interest to a broad range of neuroscientists. However, the paper is not written to be broadly accessible. Control experiments are not discussed or described, and a key prediction of the model (phase differences) is untested. The discussion, for a topic of such wide interest, is frustratingly brief to non-existent. Therefore, while I am positive about this very interesting manuscript, I would like to see the authors address some concerns in a major revision.

We very much appreciate the careful and thoughtful comments and constructive criticisms that this reviewer provided. In the revision, we made every effort to address his/her concerns and suggestions.

Major concerns

Q1: My biggest concern is essentially the lack of a control experiment. While the maturation and material changes of vocal tract properties are explored in this study for the cries → phee transition, nothing is said about the other sounds that are produced using the same apparatus, yet do not appear to acoustically change over development (twitters or trills, as discussed in Takahashi et al 2015). These sounds also appear to be of high-frequency like the phee, so similar issues would hold. On the flip side, it does not seem to be the case that adults are incapable of producing low-frequency sounds – for example, Agamaite et al 2015 describe “Egg” and “Ock” sounds that have sub-1KHz f0s. If it were the case that the control space for low-frequency production is lost in adults, how are they still able to produce these sounds?

A1: We thank the reviewer for pointing out these issues. Although perhaps we didn't make it explicit enough, the focus of our study was on the development of the contact call as this is the only call in the marmoset repertoire that undergoes significant structure changes and for which the mechanistic focus to date has been on neural explanations. Thus, in our view, the adult contact call/larynx is in fact the “control” for the developing contact call/larynx.

That said, the reviewer insightfully points out that marmosets produce a number of other calls with a wide range of fundamental frequencies and asks how our findings can account for this. We believe the mature contact “phee” call is unique in the sense that it is a very long duration call that requires a flat tonal contour. Thus, its energetic requirements are different from the other calls in the repertoire (something we've addressed in a recent study: Zhang & Ghazanfar, *PLoS Biology*, 2018). In the revised manuscript, we have added the following paragraph:

“Our study focused on the contact call development—from cries to phee—of marmoset monkeys. This is because it is the only call in their repertoire that undergoes a significant transformation in acoustic properties during the course of development⁹. Its long duration, loudness and clear tonality (in mature versions) makes them uniquely difficult, more energetically costly, to produce when compared to other calls in the species repertoire¹⁴. Thus, while marmoset monkeys have a rich repertoire of vocalizations⁶⁰, all but the contact call sound adult-like at the beginning of postnatal life⁶¹. A number of call types, such as “trills”, “trill-phees” and “twitters”, have a similar fundamental frequencies as the contact call^{2,63}. However, these other call types are all very short in duration and without a sustained, tonally flat contour (that is

initially noisy). Other call types in the species repertoire can sometimes have fundamental frequencies above 10 kHz (“tsik” calls) as well as below 1 kHz (like the “egg” calls); these calls are very short in duration⁶⁴. Our study did not attempt to account for the production of these call types; there is no reason to think that the changing material properties of the larynx prevent the production of these other call types via a different set of laryngeal control dynamics. For example, the sub-1-kHz calls may be produced when the vocalis muscles within the vocal folds are shortened, a possibility that cannot be simulated in our experimental setup. Along similar lines, human vocal folds can exhibit several stable modes of vibration with different f_0 ranges, such as chest and falsetto regime⁶⁵. We cannot exclude that multiple vibratory modes in VF or VM are possible as observed in the human larynx. For contact call development in marmosets, however, our data and neural modeling suggest that is the combination of changing laryngeal material properties (switching to vocal membranes as the sound source) and the development of neural control of respiratory and laryngeal synchronization that transforms them from long-duration, noisy cries into long duration, tonal phee calls.”

Q2: Which control parameter is responsible for the loudness of the produced sounds?

A2: We thank the reviewer for raising this issue. This is an important point we did not include in our first submission. However in our revision we calculated the source level (SL) of in vitro and in vivo vocalizations. The SL of the sounds from infant and adult larynges were plotted in the pressure-strain space in the new Figure 4. Our results show that SL was positively correlated with pressure and also with vocal fold strain. VM dominated oscillation is thus louder than VF oscillation confirming the predictions by Mergell et al (1999, cited in the manuscript). Furthermore, we estimated the source levels of in vivo vocalizations (n=10 infants and n=9 adults). Sound produced by the adult larynx was about 15 dB louder than the infant larynx. These results are consistent with the in vivo data in that (1) adult phee calls are about 12 dB louder than infant phee calls and (2) infant pee calls are about 10 dB louder than the infant cries.

Our in vitro setup furthermore allowed us to calculation the mechanical efficiency (ME) of sound production. We found that the vocal fold strain was positively correlated with mechanical efficiency and that the adult larynges were about 7 dB more efficient at the same f_0 . We thus argue that there is a benefit for the marmosets to produce high f_0 vocalizations for the sake of loudness, a way to increase their range of communication.

We added the following sections in the Results:

“Adult larynx is more vocally efficient than the infant larynx.

Our data show that the function of the VMs--a morphological innovation observed in New World monkeys, cats, and bats--is to i) increase the f_0 range and ii) decrease PTP, corroborating previous theoretical predictions^{11,31,45}. Because decreased PTP is indicative of increased vocal efficiency (leading to higher amplitude vibrations for a given lung pressure)¹¹, vocalizations could theoretically be produced louder with less energy (thereby extending their communicative range). To test this, we calculated the sound source level (SL) as a function of pressure and vocal fold (VF proper and VM) strain. We found that the SL of the emitted sound was positively correlated with pressure as well as strain in both infant and adult larynges ($p < 0.001$ for slopes of multiple linear regression; Fig. 4a). This is consistent with the prediction that the VM--the source oscillator at the higher strain levels--is more energy efficient than the VF proper. Overall, the adult larynx produced sound about 15 dB louder than the infant larynx across all sets of parameters (Fig 4a). These findings from the excised larynx preparation are consistent with what is observed in naturally produced vocalizations (Fig. 4b). The infant's mature-sounding contact calls are generally louder than its cries, and the adult contact calls are louder than infant's contact calls ($p < 0.001$, ANOVA). Assuming sound transmission loss by spherical spreading⁴⁶, the 15 dB loudness increase

of the contact call from infant to adult results in a roughly 6 times larger communicative distance. To produce louder sounds, the adult larynx may simply draw more mechanical energy or it might be more efficient in converting mechanical energy to sound. To test this, we estimated its mechanical efficiency (ME): the fraction of the aerodynamic power that was converted to acoustic power⁴⁷. We found that the adult larynx was more efficient at all parameter settings (Fig. 4c), demonstrating that ME increases over development. In addition, the ME was positively correlated with strain for both infant and adult larynges ($p < 0.001$ for slopes of multiple linear regression; Fig. 4d). Thus, it is more energy efficient to produce high f_c sounds when energetic constraints allow for them.”

Here are the methods we used for calculating efficiency:

“Efficiency analysis. To estimate the source level (SL) of the emitted sound from the excised larynx, we calculated the root mean square (RMS) of the calibrated sound pressure within 12.8 ms sliding windows. The SL at 1 meter distance was estimated using

$$SL = 20 \log_{10} \frac{p}{p_0} + 20 \log_{10} r,$$

where p is the RMS of sound pressure, $p_0 = 20 \mu\text{Pa}$ is the reference pressure and r the distance from the excised larynx to the microphone.

To calibrate the SL of marmoset vocalizations, a speaker broadcasting a constant 45 dB-SPL pink noise was placed in the room to mask occasional noises. The sound pressure of marmoset vocalization was calibrated to the background noise. A microphone was positioned at a distance of 0.76 m to the testing box where the marmoset was housed. The RMS of the calibrated sound pressure was used to calculate the SL using the above formula.

The mechanical efficiency was calculated using

$$ME = 10 \log_{10} \frac{P_{\text{acoustic}}}{P_{\text{aerodynamic}}},$$

where the acoustic power in Watts $P_{\text{acoustic}} = IA = \frac{4\pi r^2 p^2}{\rho v}$, and the aerodynamic power in Watts

$P_{\text{aerodynamic}} = P_s \dot{V}$. Here we used air density $\rho = 1.2 \text{ kg/m}^3$, speed of sound $v = 344 \text{ m/s}$, p the RMS sound pressure in Pa, P_s the subglottal pressure in Pa and \dot{V} the glottal flow rate in m^3/s ^{47,68}.”

Q3: The contact calls studied here seem extremely loud, and it appears that if one emphasizes loudness as a critical factor, you could take an underlying linear system and over-drive it to produce nonlinear outputs with harmonic distortions. Could this be an alternative model to explain why a developmental change is seen for pheeas but not other, quieter, sounds?

A3: The contact calls are loud and long, requiring more energy than other calls. Previously in Teramoto 2017, it was proposed that the cry is a low-energy solution for the infants to produce audible contact calls given their underdeveloped larynx and muscle strength. The dense harmonic stack can be benefitted from the amplification of the vocal tract. Our results in this study show that indeed the infant larynx has an enlarged area at low strain that can produce the cry-like low f_0 harmonic stack without the need to over-drive the vocal folds; this region significantly shrank in the adult larynx. The in vivo cries and pheeas we observed are all regular modal sounds that do change over development. The in vitro sounds produced were also largely regular limit cycle vibrations and overlapped with the in vivo range. So we don't need to “overdrive” into higher modes or deterministic chaos; this notion is not an alternative model to explain the changes. We agree with the reviewer that the developmental change is seen for

phoe call but not for other less energy-demanding call types; but the reason is that a low-energy strategy was adopted instead because of the high-energy cost of phoe.

Q4: Related to the authors' earlier study, how do the authors imagine an instructive signal (such as parental feedback) facilitates the transformation from cries → phoes? Is it entirely based on the synchronization of respiration and muscle tension as their model suggests? But then wouldn't marmosets also need to learn synchronization for the other sounds they produce?

A4: The synchronization is through increases in the strength of the common input to the respiratory and the laryngeal CPGs. The instructive signal can act on enhancing this common input to facilitate the transformation from cry to phoe. Phase-locked synchronous muscle recruitment is known to generate diverse motor actions. Our proposed model is a simple solution for the phase difference we observed in the muscle control of cry and phoe. The model is by nature oversimplified without taking other types of call production into consideration. To include other call types, more nonlinearity of the neural dynamics is needed to account for the spectral and temporal structures and a possible solution has been proposed in a previous work (Zhang & Ghazanfar 2018). The point is that in none of these models do the marmosets need to explicitly learn the motor actions or synchronization, as they are inherent to the dynamics of the CPGs. The learning occurs at a higher level that directs the CPG dynamics to enter different regimes and consequently shifts the vocal behavior through development.

In our revision, we addressed this issue by rewording the relevant Results section:

“Such phase-locked oscillations can emerge in the dynamics of coupled oscillators⁵⁰. To show how the synchronized laryngeal-respiratory control may arise over the course of development (perhaps, facilitated by social reinforcement^{49,51}), we generated a coupled oscillator model of laryngeal-respiratory CPGs, where their synchronization is positively correlated with their degree of coupling⁵⁰. To account for the higher order control that may coordinate vocal CPG output³, the model CPGs received a common input through which the CPGs are indirectly coupled (Fig. 5f). Within a range of parameter settings, the common input itself oscillates and forces the CPGs to oscillate at the same frequency with a fixed phase difference. The degree of coupling, negatively related to the phase difference between the CPGs, is controlled by the strength of this input (Fig. 5g). Based on this model, shifts in phase difference towards zero between pressure and strain over the first two months of life require increases in the input strength from this putative higher order area to the CPGs (Fig. 5h,i).”

Q5: In Fig. 2, it seems that the experiments were performed by holding one parameter and ramping the other parameter. Given the modeling in Fig. 4, it would be far more interesting to co-vary the parameters in-phase or out-of-phase. In an adult vocal tract, maybe it is possible to generate low frequency sounds after all by changing this phase difference. This would also go towards addressing concern (1) above.

A5: As the acoustic map of marmoset larynx has never been explored, the first experiment is to study the quasi steady state as we did. Our parameter range should cover the capacity of the larynx under full physiological range of CT muscle contractions and subglottal pressure. Thus an in-phase or an out-of-phase operation should not add more parameter combinations than what we already explored, unless the dynamics itself contributes to the acoustics due to hysteresis. We agree that co-variations would be very interesting and can potentially explain the production of other call types or provide alternative explanations for the cry and phoe

production; unfortunately we cannot do those experiments anymore. We predict that given the slow temporal dynamics (order of seconds in fig 5D) our steady state parameter space behaviors will accurately predict the VF vibratory behavior. We added the following sentence in the Methods:

“By slowly changing the VF length (sawtooth at 2 Hz), we predict that VF vibratory behavior was in a quasi-steady state.”

Q6: Please state the mean lengths of the vocal folds in infants and adults. VF length and VF strain are used interchangeably when describing Fig. 2 results, while these are of course related, it would help to consistently stick to absolute or relative units. It would be instructive to also provide, as extended data figure, the plots in 2e-g, with VF *stress* on the x-axis, as the authors mention in line 125 that this could be a critical factor in generating sounds with high f0s.

A6: We added the mean VF length to the Results:

“The vocal fold strain was calculated as the fractional change of vocal fold length with respect to the resting length. The mean resting lengths of infant and adult vocal folds were 1.4 ± 0.2 mm and 2.8 ± 0.1 mm, respectively.”

Moreover, Fig. 2d has been fixed to VF strain to keep consistent. By plotting f0s in the pressure-stress space as the reviewer suggested, the relationship between stress and f0 at the high strain region is revealed and is consistent with the string model (Figure 1 below).

We generated a pressure-stress space by converting the vocal fold strain to VM stress using the fitted model we report in the manuscript. In this way, we can compare the f0s produced across the same sets of parameters. We show that the infant larynx produced f0s around 1.5-2 times as the adult larynx at the same stress, approximately inversely proportional to their vocal fold lengths (Figure 1a). In addition, the WE of the adult map was higher almost everywhere than the infant one in the pressure-stress space as more harmonics within the same frequency range were co-occurring with the lower f0s of the adult larynx (Figure 1b). Thus, the deviation from the string model occurred only at the low strain region of the infant larynx; while at the VM dominant, high strain region, the oscillatory frequency was largely predicted by the vocal fold length.

Figure 1 | Acoustic map in pressure-stress space. **a**, Mean f_0 maps of infant ($n=3$) and adult ($n=4$) larynges in the pressure-stress space. Iso- f_0 contours (in Hz) are overlaid. **b**, Mean WE maps in the pressure-stress space. Iso-WE contours (in dB) are overlaid.

However, we did not see a reason to include this analysis in the manuscript as does not add to the main conclusions.

Q7: Why are there systematic differences in the sampling of the pressure-strain space between adults and infants? Looks like the minimum pressure applied was ~ 2 kPa for infants and ~ 0.5 kPa for adults. The reason for this is not clear from the Methods. What do the blank regions in these plots signify – data not acquired, or f_0 not quantifiable? Since the f_0 of the infant cry is ~ 500 Hz, it would be useful to mark this contour in Fig. 2e to g.

A7: There are no differences. We systematically sampled length and pressure. We only plot acoustic parameters where sound was produced. The 500 Hz contour is not possible to obtain as the low frequencies we observed were in the range of 500-1000 Hz and they were averaged out. In the Methods, we explain:

“Control space construction for acoustic maps. To systematically explore acoustic output in the control space of subglottal pressure and vocal fold strain, we subjected the larynx to a set of subglottal pressure between 0 kPa and 5 kPa with 0.5 kPa intervals while manipulating the vocal fold length from 0 to 50% strain. By slowly changing the VF length (sawtooth at 2 Hz), we predict that VF vibratory behavior was in a quasi-steady state. The strain of the vocal fold was calculated as $\epsilon(t) = (L(t) - L_0)/L_0$, where L_0 is the vocal fold length at rest. The maximal strain was set to approximately 50%. f_0 and WE were calculated on a sliding window of 12.8 ms with 10 ms overlap when sound was produced. The pressure and the vocal fold strain corresponding to the midpoint of the sliding window were identified. To compare the acoustic range between infant and adult larynges, we calculated f_0 and WE within grids of vocal fold strain (bin size=0.05) and pressure (bin size=0.5 kPa). If their vocal ranges were the same, one would expect that the acoustic

features within the same grid were the same and they should fall onto the 45° line of the infant vs. adult plots. We thus compared the slopes of those plots to 1. A slope greater than 1 means a greater range in the infant larynx than the adult's, and vice versa."

Q8: Are these systematic differences in the VF thickness between infants and adults that could explain some of the differences in the stress-strain curves in Fig. 3b for VF?

A8: Ideally, stress is independent of thickness. In our experiment, we could not guarantee that the stress is uniformly distributed within a cross-section, and so the stress of the thicker tissue will tend to be underestimated because of an overestimated effective cross-sectional area. The area of the adult VF is about 3 times as large as the infant VF. However, in the new fig. 3b where we plotted VM and VF in separate plots as this reviewer suggested, it is obvious that there is not a significant difference between the infant and adult VF curves. The difference between the VM curves exists mainly in the shape of the curves, which cannot be simply explained by scaling the size. Thus, although there is a systematic difference in the VF thickness between infants and adults, we believe it plays a minor role in explaining the differences in the curves.

The same argument also applies to the difference between the VM and VF curves whether the difference is purely from an over- or under-estimate of the size when calculating the stress. The cross-sectional area of the VF is about 3-4 (3.1 for infants and 3.8 for adults) times as large as the VM. In the worst-case scenario where the difference is completely due to size difference, the VF stress should be no more than 3-4 times greater. However, even so, the VF stress would still be lower than the VM stress. Thus, the VM/VF stress difference should be in their material properties. The effect of size may exist but is hard to assess precisely.

Q9: On a related note, I would suggest plotting VM and VF on separate plots, it is difficult to compare across adult and infant VF and VM, partly because it is scaled for VM. To my eye it seems that the infant stresses at high strains are ~50% larger than adults, and this could contribute to the steeper slopes seen in Fig. 2 (another reason to show pressure-stress axes).

A9: In the new Fig. 3b, we plotted VM and VF on separate plots so the y-axes were scaled to the data. Indeed at high strains over 0.4 the infant VF diverges from the adult VF, which could explain the increase in f_0 in Fig 2F.

Q10: The paragraph beginning line 121 could be rewritten much more clearly. If a string model were true, adults would be at an advantage for producing lower frequency sounds (longer VF, smaller non-linearity in stress-strain curves). Yet they don't, suggesting that the string models are missing an important component. Then introduce the VM.

A10: We appreciate the chance to clarify. We revised this section of the Results in the following manner:

"Vocal fold material properties are a likely candidate to induce laryngeal changes at developmental timescales, but we lack the knowledge of how VF material properties could change over time. While VF vibratory kinematics, and laryngeal acoustic output, results from the complex interplay of viscoelastic properties, fluid flow, and acoustics³³⁻³⁵, modeling the VF as a 1-dimensional vibrating string can, at a first approximation, partially explain f_0 ranges observed across a variety of species, even after accounting for body size³³. Such string models predict that f_0 is proportional to tissue stress over density

and inversely proportional to vocal fold length³³. VF stress typically increases nonlinearly with strain, which allows an extension of f_0 by two different mechanisms. First, as the maximum VF length is constrained by laryngeal geometry, nonlinearly increasing VF stress with strain allows an upward extension of f_0 range. Second, any nonlinearity in the dynamical system which sets the VF vibration kinematics can lead to multiple discrete and stable attractor states of VF vibratory patterns³⁶.

We measured the stress-strain properties of VF tissues in infant and adult larynges (See methods). A salient part of the VF geometry in some species are thin apical extensions called vocal membranes (VM)^{37,38}. Their function in vocal production remains untested experimentally but it has been suggested that they act as low mass oscillators that can vibrate almost independently of the VF proper¹, and thus support, for instance, the production of high-frequency vocalizations^{38,40}. We therefore quantified the stiffness of VF proper and VM separately. The stress-strain curves were highly nonlinear and could be fitted with combined linear and exponential models at low and high strain regions^{41,42} (Fig. 3b). The linear elastic limit was around 12-20% strain (fitted parameters summarized in Table 1). Overall, for both infant and adult larynges, as the strain increased, the VM stress increased more rapidly than the VF stress ($p < 0.001$, linear model). Following the string model, at a strain of 50%, both the infant and adult VMs can yield a 3-4 times higher f_0 than the VF. Consistent with predictions for the role of the VM, this would enable high-frequency vocalizations at a greater efficiency¹. However, it does not explain the f_0 magnitude difference between the observed immature and mature call attractors. The stress-strain responses between infant and adult larynges were significantly different between their VMs ($p < 0.001$), but not between their VFs ($p = 0.42$). Thus, the maturation of larynx is driven at least in part by a viscoelastic change in the VM. The VM trajectories (Fig. 3b) show that in low strain regions, the stress of the infant VM only starts deviating from the VF at ~17% strain, much later in the trajectory than that of the adult VM (3.9% strain; $p < 0.001$, bootstrap). The infant VM exhibited a similar elasticity as its VF at a low strain region. The divergent point around 17% also coincides with the strain where the frequency jump occurred in the phonation tests of the infant larynx. In contrast, because the stress-strain responses of the adult VM and VF diverge almost immediately; their vibratory modes become distinct very quickly.”

Q11: A better picture of the larynx showing the VF and VM would be helpful.

A11: We provided a better picture for fig. 3a where the boundary between VF and VM is clearer.

Q12: The manuscript is highly technical and hard to read. It is not written with a general audience, such as the readership of Nature Communications, in mind. The authors seem to be well within the available word limit, and I would highly recommend that they provide more narrative explanations and interpretations. The explanation for Fig. 4 is particularly dense and hard to follow.

A12: Thank you for this important suggestion. We’ve now gone through the manuscript several times to make things more clear. We’ve also divided the Results with subheadings to make the organization obvious and to indicate to the reader what to expect. We also added a completely separate and expanded Discussion section.

Additional Comments

Q13: Please explain you term the peaks of the parameter distributions in Fig. 1c - f “attractors”. This terminology is introduced in the results section without

A13: We provide a definition of “attractor” in the Results:

“We define an attractor state as a region with a high probability density in the acoustic space.”

Q14: In Fig. 1 it would be helpful to include the spectra of the cry, subharmonic, and phee with f_0 marked as in Ext. Fig. 1b.

A14: In the new Fig. 1, we included the power spectra for the phee and cry.

Q15: The red and blue colors for the lines in Fig. 2d should be cued better and defined in the legend. Maintain the same color scheme in Ext. Fig. 2 (it is currently flipped).

A15: We thank the reviewer for this oversight; we switched the red and blue color in the new Fig. 2d.

Q16: In the supplementary movies, could you mark the orientation of the cranial and caudal sides?

A16: We marked the cranial and caudal sides in the new Movie S1.

Q17: The difference referred to in line 163 is not at all clear in Movie S1.

A17: We hope that the marked movie is clearer in showing that.

Q18: Line 179 The authors state that the adult VM has increased stiffness, but from Fig. 3b it seems that the adult VM slope is actually smaller than the infant VM slope. Is this a typo?

A18: We meant that the adult VM stiffness was greater than infant VM at the low strain region. We have edited this sentence:

“The adult larynx does not have this capacity. Because of its increased vocal membrane stiffness, the adult larynx only produces one attractor state characterized by stable, higher pitched sounds.”

Q19: line 193 For infants, the two *attractors*

A19: We fixed this typo. Thanks.

Reviewers' Comments:

Reviewer #1:

Remarks to the Author:

In the revised manuscript, Zhang and colleagues carefully addressed all my concerns. I am positive that the findings will be appreciated by a broad audience. Hereby, I recommend this manuscript to be accepted for publication in Nature Communications.

Reviewer #3:

Remarks to the Author:

I thank the authors for their efforts in addressing all my comments arising from the earlier submission. They have made the manuscript significantly clearer, and it is all but ready for publication.

A couple of minor points:

1) The authors' point about the cry to phee transition being a special case because of the tonality, loudness, and length is well-taken. I appreciate the discussion section on the other vocalizations. But I would recommend that they make this point clearer in the Abstract and Introduction which still leads the reader to believe that this applies to all sounds.

I suggest editing line 25 "Here we show that vocal state changes in infant marmoset monkeys..." to "Here we show that vocal state changes in marmoset monkey contact calls, which transition from noisy cries in infants to tonal calls in adults,..."

line 63 "Over the course of approximately two months..." to "Over the course of approximately two months, infants exhibit changes in the acoustic properties of cry calls, that reflect a transition from producing mostly immature versions of the contact call (i.e., cries) to ..."

2. In Fig. 2d, the authors report a sharp transition in the f_0 from ~ 2 to ~ 9 kHz at 15% strain. But in the example in 2e as well as population data in 2f, the 2 to 9 kHz occurs over a much expanded strain range (strain 0.2 to ~ 0.45). Please clarify or use more representative example.

Response to reviewer 3

Reviewer #3 (Remarks to the Author):

I thank the authors for their efforts in addressing all my comments arising from the earlier submission. They have made the manuscript significantly clearer, and it is all but ready for publication.

A couple of minor points:

1) The authors' point about the cry to phee transition being a special case because of the tonality, loudness, and length is well-taken. I appreciate the discussion section on the other vocalizations. But I would recommend that they make this point clearer in the Abstract and Introduction which still leads the reader to believe that this applies to all sounds.

We now make clear in the abstract and introduction that we are referring to contact calls only.

I suggest editing line 25 "Here we show that vocal state changes in infant marmoset monkeys..." to "Here we show that vocal state changes in marmoset monkey contact calls, which transition from noisy cries in infants to tonal calls in adults,..."

We made the suggested edits.

line 63 "Over the course of approximately two months..." to "Over the course of approximately two months, infants exhibit changes in the acoustic properties of cry calls, that reflect a transition from producing mostly immature versions of the contact call (i.e., cries) to ..."

We made the suggested edits.

2. In Fig. 2d, the authors report a sharp transition in the f_0 from ~ 2 to ~ 9 kHz at 15% strain. But in the example in 2e as well as population data in 2f, the 2 to 9 KHz occurs over a much expanded strain range (strain 0.2 to ~ 0.45). Please clarify or use more representative example.

We rephrased the relevant statements to be more clear:

"At a **specific** VF length change corresponding to **17-45%** strain, we observed a sharp transition in the f_0 **in each individual** from ~ 2 to ~ 9 kHz, the latter corresponding well to the mature-sounding contact call (Fig. 2d left panel)."

"The VM trajectories (Fig. 3b) show that in low strain regions, the stress of the infant VM only starts deviating from the VF at $\sim 17\%$ **up to $\sim 45\%$ strain**, much later in the trajectory than that of the adult VM (3.9% strain; $p < 0.001$, bootstrap)."